# Improved Bounds for Reward-Agnostic and Reward-Free Exploration

**Oran Ridel** [1]  **Alon Cohen** [1][2]

## Abstract

We study *reward-free* and *reward-agnostic* exploration in episodic finite-horizon Markov decision processes (MDPs), where an agent explores an unknown environment without observing external rewards. Reward-free exploration aims to enable $\epsilon$-optimal policies for *any* reward revealed after exploration, while reward-agnostic exploration targets $\epsilon$-optimality for rewards drawn from a small finite class. In the *reward-agnostic setting*, Li, Yan, Chen, and Fan (2024b) achieve minimax sample complexity, but only for restrictively small accuracy parameter $\epsilon$. We propose a new algorithm that significantly relaxes the requirement on $\epsilon$. Our approach is novel and of technical interest by itself. Our algorithm employs an online learning procedure with carefully designed rewards to construct an exploration policy, which is used to gather data sufficient for accurate dynamics estimation and subsequent computation of an $\epsilon$-optimal policy once the reward is revealed. Finally, we establish a tight lower bound for *reward-free exploration*, closing the gap between known upper and lower bounds.

## 1. Introduction

Exploration is a central challenge in reinforcement learning (RL). Traditional formulations assume that the agent receives a reward signal throughout interaction with the environment, allowing it to directly optimize cumulative rewards. However, in many practical scenarios - such as robotic control, scientific experimentation, or personalized medicine, the rewards may be unknown or ill-defined during data collection phase. In such settings, it is desirable to explore the environment without a reward signal: to gather data that is sufficiently informative such that, once a downstream reward function is specified, a near-optimal policy can be computed without further exploration.

This separation between exploration and reward specification is motivated by settings in which rewards are designed after data collection, multiple tasks share the same environment dynamics, or reward feedback is costly or unavailable during interaction. By focusing on accurately estimating state-action transitions, such exploration provides task-independent generalization guarantees and enables planning without further environment interaction after the exploration phase. From a high-level theoretical perspective, this fundamental framework allows to disentangle exploration and exploitation in order to understand the intrinsic sample complexity of exploration.

In this paper, we study two types of frameworks that allow for such exploration: reward-free exploration (RFE) and reward-agnostic exploration (RAE).

**Reward-free exploration (RFE).** Reward-free exploration is a learning paradigm in which an agent interacts with an unknown environment without observing any external reward signal during the exploration phase. The idea was formalized in a modern context in recent work (Jin et al., 2020b), which demonstrated efficient sample complexity guarantees for exploration in a tabular Markov decision process (MDP). Crucially, the downstream reward may be arbitrary and is not assumed to belong to any restricted class.

Their exploration scheme applies a no-regret algorithm over a sequence of rewards defined as indicator functions of state-action visitations. This allows them to construct a "policy cover" that is later sampled from to obtain a high-quality estimate of the MDP transitions. Their method achieves a sample complexity of $\widetilde{O}(H^5|S|^2|A|/\epsilon^2)$, where $S, A, H$ and $\epsilon$ represent the state space, action space, horizon of the MDP and the desired accuracy, respectively. Subsequently, this bound was refined by (Kaufmann et al., 2021; Ménard et al., 2021; Li et al., 2024b), the latter two providing an algorithm that reaches the sample complexity of $\widetilde{O}(H^3|S|^2|A|/\epsilon^2)$ in the reward-free exploration setting.

It has been shown by Jin et al. (2020b) that the lower bound for the time-homogeneous RFE task is $\Omega(|S|^2|A|H^2/\epsilon^2)$. Li et al. (2024b) conjecture that the minimax lower bound for the time-inhomogeneous RFE task is a factor $H$ larger

[1]Department of Engineering, Tel Aviv University, Tel Aviv, Israel [2]Google Research, Tel Aviv, Israel. Correspondence to: Oran Ridel <oranridel@mail.tau.ac.il>, Alon Cohen <alonco@tauex.tau.ac.il>.

*Proceedings of the 43ʳᵈ International Conference on Machine Learning*, Seoul, South Korea. PMLR 306, 2026. Copyright 2026 by the author(s).

than the time-homogeneous case, matching the upper bound of Ménard et al. (2021). In this paper, we prove this to be the case.

**Reward-agnostic exploration (RAE).** Reward-agnostic exploration (Zhang et al., 2020) is closely related to RFE in that the agent explores an unknown MDP without access to external reward signals. The key distinction between RFE and RAE lies in that in RAE, the reward is assumed to be drawn from a predetermined class of size $\text{poly}(|S|, |A|, H)$, whereas in RFE no such restriction is imposed. Although this difference may appear mild, it has significant implications for sample complexity, saving a factor of $|S|$ compared to the sample complexity of RFE.

Indeed Li et al. (2024b) achieved an optimal sample complexity bound for RAE of $\widetilde{O}(H^3|S||A|/\epsilon^2)$, matching the lower bound for Best Policy Identification (BPI), given by Domingues et al. (2021). (In this context, BPI can be thought of as a reward-agnostic exploration where the size of the reward class is one.) However, the bound of Li et al. (2024b) suffers from a high-complexity low-order term that makes their bound optimal only for sufficiently small $\epsilon$. In our work we considerably reduce this low-order term, making our approach more widely applicable.

**Online Reinforcement Learning.** Our sample complexity improvement relies on the use of online MDP algorithms. In online reinforcement learning, an agent interacts with an environment over a sequence of episodes and must repeatedly select a policy before observing the corresponding reward function. The rewards are assumed to be generated arbitrarily, possibly non-stochastically. The learner's goal is to maximize their cumulative reward over time, and performance is typically evaluated through regret, defined as the difference between the learner's total reward and that of the fixed best policy in hindsight.

The online MDP setting was first proposed by Even-Dar et al. (2009), as an extension to the standard MDP and as a first step in bridging between reinforcement learning and adversarial online learning. In recent years, there have been several works devoted to this setting, including Rosenberg & Mansour (2019); Jin et al. (2020a); Zimin & Neu (2013), which inspired some of the principles used in our paper.

**Additional related work.** The study of reward-free exploration dates back to early approaches such as the R-max algorithm (Brafman & Tennenholtz, 2002) that incur a sample complexity on the order of $\widetilde{\Omega}(H^{11}|S|^2|A|/\epsilon^3)$ (see Jin et al., 2020b, Appendix A).

Zhang et al. (2021) considered a variant of the problem under "totally-bounded rewards," where the cumulative reward over a trajectory is at most 1. When their results are translated into the standard bounded-reward setting (i.e., each instantaneous reward is at most 1), their algorithm achieves

a sample complexity of $\widetilde{O}(H^2|S|^2|A|/\epsilon^2)$ episodes for time-homogeneous MDPs.

For reward agnostic exploration, Zhang et al. (2020) proposed a model-free algorithm that requires $\widetilde{O}(H^5|S||A|/\epsilon^2)$ samples. Parallel lines of research have focused on coverage-oriented exploration objectives (Chen et al., 2022; Al-Marjani et al., 2023), while others target pure exploration for best policy identification (Russo & Vannella, 2024; Russo & Pacchiano, 2025).

For convenience, the relevant sample complexity bounds (both high and low order terms) are arranged in Table 1.

**Our contributions.** First, we present an **algorithm** (Section 3.2) that achieves the optimal high order term complexity (up to logarithmic factors), for both reward-free and reward-agnostic setting ($\widetilde{O}(|S|^2|A|H^3/\epsilon^2)$ and $\widetilde{O}(|S||A|H^3/\epsilon^2)$ respectively), while suffering from a smaller low order term than Li et al. (2024b). While our established bound for the RFE setting is looser than the one presented by Ménard et al. (2021), it is provided here for theoretical completeness.

Our method replaces the multiple instantiations of no-regret algorithms in previous work with a single run of an online MDP algorithm over a sequence of carefully constructed reward functions. This significantly reduces the cost of the low-order sample complexity by reusing samples throughout the learning process. It is followed by estimation of the transitions and an optimized planning procedure that leverages the guarantees of our online algorithm.

Second, we prove a **tight lower bound** (Section 6) on the sample complexity of reward-free exploration for time-inhomogeneous MDPs. Our bound entails that existing upper bounds are in fact optimal (up to logarithmic factors; e.g., Ménard et al., 2021), solving the conjecture of Li et al. (2024b).

**Our focus.** The focus of this work is the theoretical analysis of finite horizon tabular MDPs. Our guiding philosophy is to understand tabular MDPs as comprehensively as possible, while living non-tabular MDPs and empirical results as an important avenue for future work.

## 2. Preliminaries

A finite-horizon time-inhomogeneous Markov Decision Process (MDP) is defined by $\mathcal{M} = (S, A, H, \{P_h\}_{h=1}^H, r)$, where $S$ is a finite state space, $A$ is a finite action space, $H$ is the planning horizon, $P : S \times A \times [H] \mapsto \Delta(S)$ is the transition kernel at step $h$, and $r : S \times A \times [H] \mapsto [0, 1]$ is the reward function.

A (Markov) policy $\pi : S \times [H] \to \Delta(A)$ specifies a distribution over actions given a state and time step. For a

*Table 1.* Comparison of sample complexity bounds for reward free, reward agnostic and best policy identification algorithms on episodic finite horizon MDPs, omitting constants and logarithmic terms.

| SETTING | PAPER | UPPER BOUND | LOWER BOUND | LOWER BOUND INHOMOGENEOUS |
|---|---|---|---|---|
| REWARD-FREE | JIN ET AL. (2020B) | $\frac{H^5|A||S|^2}{\epsilon^2} + \frac{H^7|A||S|^4}{\epsilon}$ | $\Omega\left(\frac{H^2|A||S|^2}{\epsilon^2}\right)$ | × |
| | KAUFMANN ET AL. (2021) | $\frac{H^4|A||S|^2}{\epsilon^2}$ | | |
| | MÉNARD ET AL. (2021) | $\frac{H^3|A||S|^2}{\epsilon^2}$ | | |
| | THIS PAPER | $\frac{H^3|A||S|^2}{\epsilon^2} + \frac{H^4|A||S|^2}{\epsilon}$ | $\Omega\left(\frac{H^3|A||S|^2}{\epsilon^2}\right)$ | ✓ |
| BPI | DOMINGUES ET AL. (2021) | | $\Omega\left(\frac{H^3|A||S|}{\epsilon^2}\right)$ | ✓ |
| REWARD-AGNOSTIC | ZHANG ET AL. (2020) | $\frac{H^5|S||A|}{\epsilon^2}$ | $\Omega\left(\frac{H^2|A||S|}{\epsilon^2}\right)$ | × |
| | LI ET AL. (2024B) | $\frac{H^3|A||S|}{\epsilon^2} + \frac{H^6|A|^4|S|^4}{\epsilon}$ | | |
| | THIS PAPER | $\frac{H^3|A||S|}{\epsilon^2} + \frac{H^4|A||S|^2}{\epsilon}$ | | |

policy $\pi$ and reward function $r$, the *value function* and the *Q-function* at step $h$ are defined as

$$V_h^\pi(s) := \mathbb{E}_{\pi,P}\left[\sum_{t=h}^H r_t(s_t, a_t) \,\bigg|\, s_h = s\right],$$

$$Q_h^\pi(s,a) := \mathbb{E}_{\pi,P}\left[\sum_{t=h}^H r_t(s_t, a_t) \,\bigg|\, s_h = s, a_h = a\right].$$

The optimal value functions are defined as $V_h^\star(s) := \max_{\pi \in \Pi} V_h^\pi(s)$, and the optimal policy $\pi^\star$ satisfies $V_h^{\pi^\star} \equiv V_h^\star$. We also write $Q_h^\star(s,a) := Q_h^{\pi^\star}(s,a)$. Moreover, by the Bellman optimality criterion (Puterman, 2014): $V_h^\star(s) = \max_{a \in A} Q_h^\star(s,a)$. As a notation, define $[P_h V_{h+1}](s,a) := \mathbb{E}_{s' \sim P(\cdot|s,a)} V_{h+1}(s')$ and $P_{h,s,a} := P_h(\cdot \mid s,a) \in \Delta(S)$.

For a policy $\pi$ and transition dynamics $P$, the *occupancy measure* at step $h$ is defined as (see Puterman, 2014):

$$\mu_h^{\pi,P}(s,a) := \mathbb{P}[s_h = s, a_h = a \mid \pi, P].$$

Intuitively, $\mu_h^{\pi,P}(s,a)$ represents the probability of visiting the state-action $(s,a)$ at time step $h$. The expected return of policy $\pi$ is linear in its occupancy measure:

$$\mathbb{E}_{\pi,P}\left[\sum_{h=1}^H r_h(s_h, a_h)\right] = \sum_{s,a,h} \mu_h^{\pi,P}(s,a) r_h(s,a).$$

In addition, note that $\mu_1(s) := \sum_a \mu_1^{\pi,P}(s,a)$ represents the initial state distribution. Any valid occupancy measure satisfies the following constraints:

$$\sum_{s,a} \mu_h(s,a) = 1, \forall h \in [H] \tag{1}$$

$$\sum_{s,a} \mu_h(s,a) P(s' \mid s,a) = \sum_a \mu_{h+1}(s',a), \tag{2}$$

$$\forall h \in [H-1], s' \in S,$$

and any function $\mu : S \times A \times [H] \mapsto [0,1]$ satisfying Eqs. (1) and (2) induces a policy given by

$$\pi_h^\mu(a \mid s) = \frac{\mu_h(s,a)}{\sum_{a' \in A} \mu_h(s,a')}. \tag{3}$$

Indeed, this mapping between occupancy measures and Markov policies is a bijection. We denote by $\Lambda \subseteq [0,1]^{|S| \times |A| \times H}$ the convex set of all valid occupancy measures, i.e., satisfying Eqs. (1) and (2).

**Online Mirror Descent.** OMD (see, e.g., Hazan et al., 2016) is a general framework for online optimization that extends gradient-based methods by incorporating problem-specific geometry. We focus on online linear optimization over a convex set $\Lambda \subseteq \mathbb{R}^d$ where the goal is to minimize the regret compared to an arbitrary sequence of reward vectors $r^1, \ldots, r^T$ defined as $\max_{\mu \in \Lambda} \sum_{t=1}^T \langle r^t, \mu - \mu^t \rangle$. OMD[1] starts from some $\mu^1 \in \Lambda$, and receives a step size parameter $\eta > 0$. It repeatedly updates $\mu^{t+1} = \arg\max_{\mu \in \Lambda}\{\eta\langle\mu, r^t\rangle - D_R(\mu \| \mu_t)\}$, where convex differentiable $R$ is a regularization function, and $D_R$ is the Bregman divergence defined as $D_R(x\|y) := R(x) - R(y) - \langle\nabla R(y), x - y\rangle$. We have the following guarantee for OMD:

**Theorem 2.1.** *For any $u \in \Lambda$, and any $r^1, \ldots, r^T \in \mathbb{R}^d$, OMD guarantees:*

$$\sum_{t=1}^T \langle r^t, u - \mu^t \rangle \leq \frac{1}{\eta} D_R(u\|\mu^1) + \frac{1}{\eta}\sum_{t=1}^T D_R(\mu^t\|\mu^{t+1}).$$

## 3. Problem Formulation and Main Results

We consider a finite-horizon time-inhomogeneous tabular MDP. A learner is given the parameters of the MDP: $S, A, H$

---

[1]In our work we in fact use Online Mirror *Ascent*, however we keep the naming OMD for consistency with previous work.

and an accuracy parameter $\epsilon > 0$. They then interact with the MDP for $K$ episodes, where $K$ is chosen by the learner given the aforementioned parameters.

At the beginning of episode $k = 1, \ldots, K$, the learner selects a policy $\pi_k$ and interacts with the environment for $H$ steps, starting from an initial state $s_1 \sim \mu_1$. The learner observes the resulting trajectory $\{(s_h, a_h)\}_{h=1}^H$, consisting of the state-action pairs encountered during the episode. Crucially, the learner never observes the rewards of the MDP, and the intrinsic reward at episode $k$ depends only on the data collected up to episode $k - 1$. However, in the RAE setting, the learner knows the predefined class to which the rewards belong to.

After the $K$ episodes the learner can no longer interact with the MDP, and subsequently the true reward is revealed. The agent's goal, both for reward-free and reward-agnostic exploration, is to return an $\epsilon$-optimal policy $\hat{\pi}$, namely

$$\mathbb{E}_{s \sim \mu_1(s)}[V_1^\star(s) - V_1^{\hat{\pi}}(s)] \leq \epsilon.$$

The sample complexity of the problem is the minimum $K$ that can attain this guarantee.

**Note.** We use union bounds over $|S|$, $|A|$, $H$, and $T$ throughout. For brevity, we define the logarithmic factor $\ell_0 := \ln\left(\frac{|S||A|H}{\delta\varepsilon}\right)$, which captures the worst-case contribution from these bounds.

### 3.1. Main results

Our first result pertains to *reward-agnostic exploration*. We prove a sample complexity bound whose high-order term (in terms of $\epsilon$) is optimal up to logarithmic factors, and its low order term is a significant improvement over Li et al. (2024b). (See Table 1 for comparison of upper and lower bounds.) In Section 3.2, we present a high-level description of our algorithm and explain our technical contributions.

**Proposition 3.1.** *For a reward function $r$ drawn from a class of size $\mathrm{poly}(|S|, |A|, H)$, for any $\epsilon > 0$ and any $\delta \in (0, 1)$, w.p. at least $1 - \delta$, the number of samples required for Algorithm 1 to return an $\epsilon$-optimal policy is*

$$O\left(\frac{|S||A|H^3\ell_0^3}{\epsilon^2} + \frac{|S|^2|A|H^4\ell_0^3}{\epsilon}\right).$$

The proof of the proposition is divided between Sections 4 and 5.

For our second result, in Section 6 we present a tight lower bound on the sample complexity of *reward-free exploration* in time-inhomogeneous episodic MDP, solving the conjecture of Li et al. (2024b). In particular, this result highlights the optimality of prior work (e.g., Ménard et al., 2021).

**Proposition 3.2.** *Let $C > 0$ be a universal constant, $|A| \geq 2$, $|S| \geq C\log_2 |A|$ and $H \geq C\log_2 |S|$. Any reward-free exploration algorithm Alg that returns an $\epsilon$-optimal*

*policy for any reward function, with confidence level $1/2$ and accuracy parameter $\epsilon \leq \min\{1/5, H/48\}$ must collect at least*

$$\Omega\left(\frac{|S|^2|A|H^3}{\epsilon^2}\right)$$

*trajectories in expectation.*

### 3.2. Algorithm Overview

---

**Algorithm 1** META-ALGORITHM

---

1: **input:** Accuracy parameter $\epsilon$, confidence $\delta$, state space $S$, action space $A$, horizon $H$.
2: **create** exploration policy (Algorithm 2):
    $\pi_{\mathrm{exp}} \leftarrow$ EXPPOLICYCREATION$(S, A, H, \epsilon, \delta)$.
3: **estimate** dynamics (Algorithm 3):
    $\mathcal{D}, \hat{P} \leftarrow$ DYNAMICSEST$(\pi_{\mathrm{exp}}, S, A, H, \epsilon, \delta)$.
4: **observe** reward function $r$.
5: **compute** near-optimal policy (Algorithm 4):
    $\hat{\pi} \leftarrow$ POLICYEST$(r, \mathcal{D}, \hat{P})$.
6: **return** $\hat{\pi}$.

---

In this section, we present a high-level description of the proposed reward-agnostic / reward-free exploration algorithm. The algorithm consists of a three-stage procedure that, with probability at least $1 - \delta$, outputs an $\epsilon$-optimal policy for any reward function drawn from a predefined set of size $\mathrm{poly}(|S|, |A|, H)$ (Theorem 5.1), or for *any* reward function in the fully reward-free setting (Theorem 5.2).

We next describe the main steps of our algorithm. The correctness of these steps is shown in Sections 4 and 5 where we also prove Proposition 3.1.

**Step 1: Exploration policy creation.** This step is where the main sample complexity improvement stems from. It is carried out by Algorithm 2, which is described and analyzed in Section 4. For this step, prior work such as Jin et al. (2020b); Li et al. (2024b) used repeated applications of stochastic no-regret algorithms (e.g., EULER; Zanette & Brunskill, 2019) on carefully constructed reward functions. Each application of the algorithm essentially required re-estimating the transition function incurring additional polynomial factors of $|S|, |A|, H$. Our technique, on the other hand, is significantly more sample-efficient. It requires a single application of an online MDP algorithm, adapted from Rosenberg & Mansour (2019). We run the algorithm on a sequence of reward functions that enables a unified exploration of the entire MDP, up to some hard-to-reach states (see Section 4 for exact definition). In the end we return an average of the policies visited by the online algorithm, which forms our exploration policy for the second step of the meta-algorithm.

In Section 3.3 we explain how this phase approximately-solves a convex optimization problem over occupancy measures, which forms our high-level motivation for this step.

**Step 2: Dynamics estimation.** We estimate the transitions of the MDP using the exploration policy returned by step 1, by repeatedly generating trajectories from it. See Algorithm 3 in Section 5. The number of sampled trajectories is fixed for RFE, or depends on the size of the reward class in the case of RAE.

**Step 3: Policy estimation.** The final step focuses on computing an $\epsilon$-optimal policy given a reward function $r$ and an estimated transition kernel $\hat{P}$. We follow the approach of Li et al. (2024b) using *pessimism in the face of uncertainty* (Algorithm 4 in Section 5). This allows to bound the approximation error in terms of the quality of estimate of the expected return of the optimal policy $\pi^\star$ associated with the reward $r$. This in turn is bounded by virtue of the exploration policy returned by step 1, which guarantees uniform exploration of the state-action space. To this end, for each $(s, a, h) \in S \times A \times [H]$, we introduce a penalty term $b_h(s, a)$, derived from Bernstein-style concentration inequalities. We formalize an empirical MDP over the estimated dynamics $\hat{P}$ and rewards $r$ from which we subtract the penalty term. We then return $\hat{\pi}$ as the optimal policy in the empirical MDP.

### 3.3. Main idea: convex optimization perspective

When it comes to pure exploration tasks, with the goal of uniform transition dynamics estimation, the existence of a single Markov policy that enables such uniform exploration is proved. Indeed, consider the following convex optimization problem over all valid occupancy measures:

$$\mu^\star = \arg\min_{\mu \in \Lambda}\left\{ -\sum_{s,a,h} \log \mu_h(s, a) \right\}. \qquad (4)$$

By first-order optimality conditions of convex programs (see, e.g., Boyd & Vandenberghe, 2004), we immediately obtain:

$$\sum_{s,a,h} \frac{\mu_h^\pi(s, a)}{\mu_h^\star(s, a)} \leq |S||A|H, \ \forall \pi. \qquad (5)$$

Thus we can sample $N$ trajectories from the policy associated with $\mu^\star$, use them to estimate the transition function, and construct an empirical MDP using the estimated dynamics. We denote by $N_h(s, a)$ the number of samples obtained from state-action $s, a$ at time $h$. For any policy $\pi$, we look at the difference in its value functions between the true and the empirical MDP. Due to standard concentration inequalities argument, and an application of the value difference lemma (e.g., Jin et al., 2020b):

$$|V^\pi(s) - \hat{V}^\pi(s)| \lesssim \sum_{s,a,h} \mu_h^\pi(s, a) \sqrt{\frac{1}{N_h(s, a)}}$$

$$\approx \sum_{s,a,h} \mu_h^\pi(s, a) \sqrt{\frac{1}{N \mu_h^\star(s, a)}}$$

$$\leq \sqrt{\sum_{s,a,h} \mu_h^\pi(s, a)} \sqrt{\frac{1}{N} \sum_{s,a,h} \frac{\mu_h^\pi(s, a)}{\mu_h^\star(s, a)}}$$

$$\leq \sqrt{\frac{|S||A|H^2}{N}},$$

where the second-to-last inequality is by Cauchy-Schwarz, and the last inequality is because $\sum_{s,a,h} \mu_h^\pi(s, a) = H$ and by Eq. (5). This implies $N = \Omega(H^2|S||A|/\epsilon^2)$ suffices.

Prior work (Hazan et al., 2019) consider minimizing the negative entropy of the occupancy measures: $f(\mu) = \mathbb{E}_{s\sim\mu^\pi} \log \mu^\pi(s)$. Unfortunately, Jin et al. (2020b) show that using the maximum-entropy for exploration requires sample complexity that scales with $|S|^5$. Focusing back on Eq. (4), minimizing it in practice without knowing the transitions is a challenging task. Li et al. (2024b) consider a Frank-Wolfe-style algorithm, where each iteration involves running a no-regret algorithm. However, their method requires them to perturb the log function to make sure it does not diverge near zero and that the gradients of the function are Lipschitz. This in turn causes the iteration complexity of Frank-Wolfe to incur high-degree polynomial factors in $|S|, |A|, H$.

In the next lemma we show that Algorithm 2, used in step 1 of Algorithm 1, returns an approximate solution to Eq. (4) even though it does not optimize it directly. Moreover, we show in Section 4 that the use of an online algorithm rather than Frank-Wolfe significantly reduces the sample complexity of this step.

**Lemma 3.3** (Informal). *Let $f(\mu) = -\sum_{s,a,h} \log \mu_h(s, a)$. Let $\mu^T$ be an occupancy measure generated by Algorithm 2 that satisfies Theorem 4.2. Then*

$$f(\mu^T) - f(\mu^\star) = O(|S||A|H \log T).$$

The Lemma is loosely attained by Lemma 4.3 and the inequality $\ln(x) \leq x - 1$:

$$f(\mu^T) - f(\mu^\star) = \sum_{s,a,h} \log \frac{\mu_h^\star(s, a)}{\mu_h^T(s, a)}$$

$$\leq \sum_{s,a,h} \left[ \frac{\mu_h^\star(s, a)}{\mu_h^T(s, a)} - 1 \right] = O(|S||A|H \log T).$$

This section motivates exploration through the optimal occupancy measure $\mu^\star$, defined as the minimizer of the log-barrier objective Eq. (4). In Section 4, we leverage the reward sequence $\{\lambda^t\}_{t=1}^T$ (see Lemma 4.5), which we use to approximate $\nabla_{\mu_h(s,a)} f(\mu) = -\frac{1}{\mu_h(s,a)} \approx -\sum_{t=1}^T \lambda_h^t(s, a)$, to show that the occupancy measure $\mu^T$ produced by Algorithm 2 approximately minimizes the same objective (Lemma 3.3). Consequently, $\mu^T$ enjoys a similar coverage property up to logarithmic factors (Theorem 4.2).

# 4. Exploration policy creation

---

**Algorithm 2** CREATE EXPLORATION POLICY

---

1: **input:** state space $S$, action space $A$, horizon $H$, accuracy $\epsilon$, confidence $\delta$.
2: **init:** $N_h^1(s,a) \leftarrow 0$, $\pi_h^1(a|s) \leftarrow \frac{1}{|A|}$, $\forall(s,a,h)$, $c$ and $T$ as in Corollary 4.6.
3: **for** $t = 1, ..., T$ **do**
4:   **traverse** trajectory $(s_1^t, a_1^t, s_2^t, a_2^t, ..., s_{H+1}^t)$ using policy $\pi^t$.
5:   $\lambda_h^t(s,a) \leftarrow \frac{1}{c+N_h^t(s,a)}$, $\forall s,a,h$.
6:   $\pi^{t+1} \leftarrow$ EXPLORATION-POLICY-ALGO$(\lambda^t)$ (Algorithm 5).
7:   $N_h^{t+1}(s,a) \leftarrow N_h^t(s,a) + \mathbb{I}\{s_h^t = s, a_h^t = a\}$, $\forall s,a,h$.
8: **end for**
9: **return** $\frac{1}{T} \sum_{t=1}^T \pi^t$.

---

Algorithm 2 implements step 1 in our meta-algorithm (Algorithm 1). The algorithm generates a sequence of auxiliary reward functions $\{\lambda^t\}_{t=1}^T$ and feeds them to our online MDP algorithm EXPLORATION-POLICY-ALGO (Algorithm 5). The reward sequence is chosen in order to steer the algorithm towards policies that explore less-visited states-action pairs.

The constant $c$ serves three purposes. First, it ensures that the rewards are bounded by $1/c$ which is a requirement of the online algorithm. Second, it allows us to use a high-probability argument and approximate $c + N_h^t(s,a) \approx c + \sum_{\tau=1}^{t-1} \mu_h^{\pi^\tau}(s,a)$ up to a multiplicative constant. Third, it accounts for state-action pairs that are visited with low probability, and thus are hard to explore.

Indeed, not every $s, a, h$ triple must be estimated equally well in order to obtain an $\epsilon$-optimal policy. If a triple is visited with very small probability under *any* policy, then even a large transition estimation error there has little effect on the value function. This motivates splitting the space into *significant* and *non-significant* triples: significant ones must be explored accurately, while the rest collectively have limited impact on performance.

**Definition 4.1.** The triple $(s,a,h)$ is $\omega$-significant with respect to policy $\pi$, if $\mu_h^\pi(s,a) > \omega$. Formally, $\psi^\pi := \{(s,a,h) \mid \mu_h^\pi(s,a) > \omega\}$ is the set of $\omega$-significant $(s,a,h)$ with respect to policy $\pi$.

We are now ready for the main guarantee of Algorithm 2.

**Theorem 4.2.** *Let $\pi_{exp}$ be the policy returned by Algorithm 2, and $\psi$ be the set of $\omega$-significant $s, a, h$ w.r.t. $\pi_{exp}$. Let $\mu^T$ denote the occupancy measure associated with $\pi_{exp}$. For $c = 4|S|H^2\ell_0$, $\omega = O\left(\frac{\epsilon}{|S||A|H^2\ell_0^2}\right)$ and $T \geq 2c/\omega =$*

$O(|S|^2|A|H^4\ell_0^3/\epsilon)$, *w.p. at least $1 - \delta$ for all policies $\pi$:*

$$\sum_{(s,a,h)\in\psi} \frac{\mu_h^\pi(s,a)}{\mu_h^T(s,a)} \leq O(|S||A|H\ell_0^2), \quad \text{and}$$

$$\sum_{(s,a,h)\notin\psi} \mu_h^\pi(s,a) \leq O(\omega|S||A|H\ell_0^2).$$

The theorem shows that the exploration policy $\pi_{exp}$ provides near-uniform coverage over all *significant* triples: for *any* policy $\pi$, the occupancy ratio $\mu_h^\pi(s,a)/\mu_h^{\pi_{exp}}(s,a)$ is bounded on this set. Moreover, the cumulative probability of visiting nonsignificant triplets (w.r.t. $\pi_{exp}$) is bounded for *any* policy $\pi$. Thus, the data collected by $\pi_{exp}$ is sufficient to accurately estimate the dynamics for all policies.

**Analysis and proof sketch.** The proof of Theorem 4.2 has two parts. Lemma 4.3 gives the guarantee for Algorithm 2 with respect to *any* no-regret algorithm, and Corollary 4.6 bounds the regret of our specific choice (Algorithm 5 in Section A.2). We define the regret of the online algorithm as

$$\mathbf{Reg}_{EPA}(T) = \max_\mu \sum_{t=1}^T \langle \lambda^t, \mu \rangle - \sum_{t=1}^T \langle \lambda^t, \mu^t \rangle. \quad (6)$$

**Lemma 4.3.** *Let EXPLORATION-POLICY-ALGO be an online MDP algorithm, and let $\mathbf{Reg}_{EPA}(T)$ denote its regret after $T$ (as defined in Eq. (6)). Let $\pi_{exp}$ be the policy returned by Algorithm 2 and assume $T \geq \frac{2c}{\omega}$. Then w.p. at least $1 - \delta/2$, $\forall\pi$:*

$$\sum_{(s,a,h)\in\psi} \frac{\mu_h^\pi(s,a)}{\mu_h^T(s,a)} \leq \mathbf{Bound}_{alg1}, \quad \text{and}$$

$$\sum_{(s,a,h)\notin\psi} \mu_h^\pi(s,a) \leq \omega \cdot \mathbf{Bound}_{alg1},$$

*where $\mathbf{Bound}_{alg1} := 3\mathbf{Reg}_{EPA}(T) + 10|S||A|H \log T$.*

In order to prove Lemma 4.3, we lower bound the expression $\max_\mu \sum_{t=1}^T \langle \lambda^t, \mu \rangle$, and upper bound $\sum_{t=1}^T \langle \lambda^t, \mu^t \rangle$. This is done using the following two lemmas:

**Lemma 4.4** (Best policy cumulative reward lower bound)**.** *Let $\pi_{exp}$ be the policy returned by Algorithm 2, and $\psi$ be the set of $\omega$-significant $s, a, h$ w.r.t. $\pi_{exp}$. Let $\mu^T$ denote the occupancy measure associated with $\pi_{exp}$. w.p. at least $1 - \frac{\delta}{4}$ for all policies $\pi$:*

$$\max_\mu \sum_{t=1}^T \langle \mu, \lambda_t \rangle \geq \sum_{(s,a,h)\in\psi} \frac{2\mu_h^\pi(s,a)}{5\mu_h^T(s,a)}, \quad \text{and}$$

$$\max_\mu \sum_{t=1}^T \langle \mu, \lambda_t \rangle \geq \sum_{(s,a,h)\notin\psi} \frac{2\mu_h^\pi(s,a)}{5\omega}.$$

The proof of the Lemma combines the monotonicity of our rewards sequence $\{\lambda_t\}_{t=1}^T$, with a high-probability argument to upper bound $N_h^{T+1}(s,a)$ by either $\omega$ or $\mu_h^T(s,a)$. See proof in Section A.3.1.

**Lemma 4.5** (Learner cumulative reward upper bound). *Let* $\lambda_h^t(s,a) = \frac{1}{c+N_h^t(s,a)}$ *and* $\mu^t$ *be a reward function and an occupancy measure at episode $t$ in Algorithm 2. If we set* $\mathcal{G}(T) = 4|S||A|H\ln T$, *w.p. at least* $1 - \frac{\delta}{4}$:

$$\sum_{t=1}^T \langle \mu^t, \lambda^t \rangle \leq \mathcal{G}(T).$$

The proof of this Lemma is by combining a lower bounding $N_h^{T+1}(s,a)$ by $\mu_h^T(s,a)$ with high-probability, together with an upper bound on the resulting self-normalized sum. See Section A.3.2 for proof. Using the Lemmas above, we prove Lemma 4.3 by placing the lower and upper bounds given in Lemmas 4.4 and 4.5 and applying a union bound.

The online algorithm we employ (Algorithm 5) is based on the framework of Rosenberg & Mansour (2019). Our variant incorporates a Bernstein-type confidence set (Eqs. (11) and (12)), in the spirit of Jin et al. (2020a), and performs optimization via online mirror descent over the convex set of valid occupancy measures (Eq. (14)), denoted by $\Lambda(\mathcal{P}^t)$, where $\mathcal{P}^t$ is the confidence set at time $t$. A detailed description of the resulting optimization problem, the algorithmic procedure, and the corresponding analysis is provided in Section A.2.

In the next corollary, we state the regret bound of Algorithm 5 in the context of Lemma 4.3.

**Corollary 4.6.** *Let* $c = 4|S|H^2\ell_0$, $\omega = O\left(\frac{\epsilon}{|S||A|H^2\ell_0^2}\right)$ *and* $\mathcal{G}(T)$ *be defined as in Lemma 4.5. In addition, set* $T \geq 2c/\omega = O\left(\frac{|S|^2|A|H^4\ell_0^3}{\epsilon}\right)$ *and* $B = 1/c$. *Using Lemma A.2, w.p. at least* $1 - \delta/2$ *we directly get the following regret for our specific use case of Algorithm 5:*

$$\boldsymbol{Reg}_{EPA}(T) \leq O(|S||A|H\ell_0^2).$$

## 5. Dynamics and Policy Estimation

This section pertains to steps 2 and 3 in Algorithm 1. Step 2 (Algorithm 3) estimates the model dynamics using the exploration policy returned by step 1. The algorithm samples $N$ trajectories using the exploration policy (lines 3-6). Thereafter (lines 8-12), the transition probabilities are estimated based on the collected trajectories.

In standard offline reinforcement learning settings, evaluating a batch dataset $D$ of episodic trajectories introduces a critical analytical challenge: transitions within the same trajectory exhibit strong temporal correlations. To circumvent this, we utilize the *two-fold subsampling* framework by

(Li et al., 2024a). Their approach induces statistical independence by randomly splitting the dataset into a main set ($D^{\text{main}}$) and an auxiliary set ($D^{\text{aux}}$). By leveraging $D^{\text{aux}}$ to compute a high-probability lower bound $N_h^{\text{trim}}(s)$ on the state-visitation counts in $D^{\text{main}}$, one can draw exactly $N_h^{\text{trim}}(s)$ samples from $D^{\text{main}}$. The resulting dataset, $D^{\text{trim}}$, removes the Markovian dependencies between steps, yielding a collection of independent transitions that drastically simplifies downstream theoretical guarantees.

---

**Algorithm 3** ESTIMATE DYNAMICS
---

1: **input:** # of trajectories $N$, Exploration policy $\pi_{\text{exp}}$.
2: **init**: $\mathcal{D} \leftarrow \emptyset$.
3: **for** $n = 1, ..., N$ **do**
4:     **play** $\pi_{\text{exp}}$ and observe $z_n = (s_1^n, a_1^n, \ldots, s_{H+1}^n)$.
5:     $\mathcal{D} \leftarrow \mathcal{D} \cup z_n$.
6: **end for**
7: $\mathcal{D}^{\text{trim}} \leftarrow$ TWO-FOLD-SUBSAMPLING($\mathcal{D}$) (Li et al. 2024a, Algorithm 3)
8: **for all** $(s, a, s', h) \in S \times A \times S \times [H]$ **do**
9:     $N_h^{\text{trim}}(s, a, s') \leftarrow \sum_{n=1}^N \mathbb{I}[s_h^n = s, a_h^n = a, s_{h+1}^n = s']$.
10:     $N_h^{\text{trim}}(s, a) \leftarrow \sum_{s' \in S} N_h^{\text{trim}}(s, a, s')$.
11:     **calculate** $\hat{P}_h(s' \mid s, a) = \frac{N_h^{\text{trim}}(s,a,s')}{\max\{1, N_h^{\text{trim}}(s,a)\}}$.
12: **end for**
13: **return** $\mathcal{D}, \{\hat{P}_h(\cdot \mid s, a)\}_{s,a,h}$.

---

Step 3 is executed after the reward function is revealed. It is performed using Algorithm 4, and is based on Li et al. (2024b, Algorithm 4). The algorithm essentially performs offline reinforcement learning using the data collected by Algorithm 3. To this end, we apply *pessimism in the face of uncertainty*, whereby $Q$-value estimates are conservatively biased to account for finite data coverage. Formally, for each state-action-step triple $(s, a, h)$, we define the pessimistic $Q$-value estimate

$$\hat{Q}_h(s,a) = \max\{r_h(s,a) + \hat{P}_{s,a,h}\hat{V}_{h+1}(s,a) - b_h(s,a),\ 0\},$$
(7)

where $b_h(s,a)$ is a nonnegative penalty that quantifies uncertainty in the empirical estimate. In our algorithm, $b_h(s,a)$ is chosen based on a Bernstein-type concentration bound (see Eqs. (15) and (16) in Section B depending on RAE or RFE setting) that scales with both the empirical variance and visit counts for $(s,a)$, yielding a data-dependent lower confidence bound on $Q_h(s,a)$. This construction ensures that poorly covered state-action pairs incur larger penalties, thereby discouraging the learned policy from visiting them.

The following guarantees are made for both the RFE and RAE settings

**Theorem 5.1** (Reward-Agnostic guarantee). *Let* $\epsilon > 0$, $\delta \in (0, 1)$. *Given a reward function $r$ taken from a set of size*

**Algorithm 4** PESSIMISTIC MODEL-BASED OFFLINE RL

1: **input:** Reward function $r$, data-set $\mathcal{D}$, dynamic estimation $\hat{P}$.
2: **Init:** $\hat{V}_{H+1} = 0$.
3: **for** $h = H, ..., 1$ **do**
4:     **for** $s \in S, a \in A$ **do**
5:         **compute** $b_h(s, a)$ according to Eq. (15) for RAE and Eq. (16) for RFE.
6:         **calculate** $\hat{Q}_h(s, a)$ according to Eq. (7).
7:     **end for**
8:     **for** $s \in S$ **do**
9:         $\hat{V}_h(s) = \max_a \hat{Q}_h(s, a)$
10:        $\hat{\pi}_h = \arg\max_a \hat{Q}_h(s, a)$
11:    **end for**
12: **end for**
13: **return** $\hat{\pi} := \{\hat{\pi}_h\}_{h=1}^{H}$

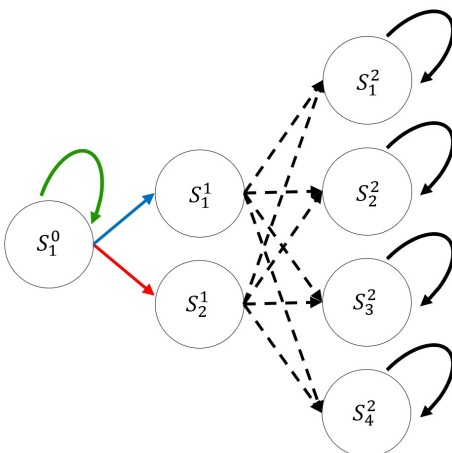

*Figure 1.* Multiple states MDP construction for lower bound. Solid lines represent deterministic transition, and dashed lines represent probabilistic transitions. Blue, red and green represent classes of deterministic actions (see Definition C.3).

$\mathrm{poly}(H, |S|, |A|)$ *of possible reward functions, a data-set* $\mathcal{D}$ *and the dynamic estimation from running Algorithm 3 for* $N = O(|S||A|H^3\ell_0^3/\epsilon^2)$ *episodes, if we run Algorithm 4, then with probability at least* $1 - \delta$

$$\mathbb{E}_{s\sim\mathbb{P}_1}\{V_1^\star(s) - V_1^{\hat{\pi}}(s)\} \le \epsilon.$$

**Theorem 5.2** (Reward-Free guarantee). *Let* $\epsilon > 0$, $\delta \in (0, 1)$. *Given any reward function* $r$, *a data-set* $\mathcal{D}$ *and the dynamic estimation from running Algorithm 3 for* $N = O(|S|^2|A|H^3\ell_0^3/\epsilon^2)$ *episodes, if we run Algorithm 4, then with probability at least* $1 - \delta$

$$\mathbb{E}_{s\sim\mathbb{P}_1}\{V_1^\star(s) - V_1^{\hat{\pi}}(s)\} \le \epsilon.$$

We now provide some intuition to the proofs of the theorems. The complete proofs are deferred to Sections B.2 and B.4 respectively. First, an important property of the pessimistic approach is that the suboptimality of $\hat{\pi}$ is bounded by the uncertainty in the value estimate of the optimal policy $\pi^\star$. We upper bound this estimate using the fact that our exploration policy guarantees near-uniform exploration over significant state-action pairs (see Theorem 4.2 and discussion in Section 3.3). For non-significant state-actions, we upper bound their contribution trivially using the guarantee from Theorem 4.2.

## 6. Lower bound for reward-free exploration

This section relates to Proposition 3.2. We prove that *any* RFE algorithm requires $\Omega(\frac{|S|^2|A|H^3}{\epsilon^2})$ samples in the worst case. To the best of our knowledge, this is the first tight lower bound established for time-inhomogeneous RFE which proves the optimality of algorithms from prior work (e.g., Ménard et al., 2021). The full analysis is found in Section C.

Previously, Jin et al. (2020b) proved a lower bound in the *time-homogeneous* setting of $\Omega(|S|^2|A|H^2/\epsilon^2)$. Our lower bound builds on their construction and extends it to the *time-inhomogeneous* setting. Their original construction was based on a primitive MDP named the "single-state construction", that consists of a single initial state connected to $2n$ absorbing states via an almost uniform transition distribution. They prove that learning this MDP in the RFE setting requires a sample complexity of $\Omega(|S||A|/\epsilon^2)$ (Jin et al., 2020b, Lemma D.2). They then build their "multi-state construction" by connecting $n$ such primitive MDPs (from the single-state construction) as the leaves of a binary tree of depth $\log_2 n$. The transitions up to the leaves are deterministic and known to the agent, allowing the agent to traverse to one of the leaves. This binary tree construction by itself, adds a factor of $\Omega(|S|)$ to the single-state lower bound. The bound is then further improved by creating a chain of rewards at each absorbing state, forcing the effective required accuracy to be $\epsilon/H$, thus adding a factor of $O(H^2)$ to the bound. All in all, this construction yields a sample complexity of $\Omega(|S|^2|A|H^2/\epsilon^2)$ (Jin et al., 2020b, Theorem 4.1).

We introduce two key modifications of the aforementioned construction (Fig. 1). First, we allow the learner to remain in the initial state via a deterministic action. Second, we extend the transition dynamics and the reward function to be time-dependent, which essentially results in $\Omega(n \cdot H)$ single-state constructions that the learner needs to learn. Since after exploration the learner must obtain a near-optimal policy for any reward function, they must also do so against a family of carefully constructed rewards that enforce that the optimal policy exits the initial state at a specific time step $h'$ and reaches a prespecified leaf. Any policy that leaves the initial state at a time step different from $h'$ incurs a loss

of at least 1. By combining this reward construction with time-dependent transition dynamics, and by allowing $h'$ to range up to $O(H)$, we introduce an additional factor of $H$ in the lower bound compared to the time-homogeneous setting of Jin et al. (2020b), thus requiring an extra factor of $H$ in the sample complexity.

## 7. Conclusion

This work advances the understanding of exploration without rewards by clarifying both algorithmic and information-theoretic limits in episodic finite-horizon MDPs. By replacing multiple independent no-regret subroutines with a single online MDP procedure operating over carefully designed reward sequences, we have shown that it is possible to reuse data more efficiently across exploration objectives. Our sample complexity gains imply that the minimax sample complexity can be realized without imposing overly restrictive conditions on the accuracy parameter $\epsilon$, making reward-agnostic exploration practical even in moderate-accuracy regimes. While this is an improvement over prior work, there is still a gap compared to the best known lower bound in low-accuracy regimes.

Moreover, our tight lower bound for reward-free exploration in time-inhomogeneous MDPs resolves a standing conjecture (Li et al., 2024b). Together with existing upper bounds, this establishes a complete minimax characterization of the sample complexity of reward-free exploration up to logarithmic factors.

## Impact Statement

This work makes theoretical contributions to understanding exploration without reward feedback in reinforcement learning, by improving sample complexity bounds and closing gaps between known upper and lower bounds for reward-free and reward-agnostic exploration. Advancing these foundational aspects deepens our understanding of the intrinsic difficulty of exploration in sequential decision-making problems and may inform future algorithm design in areas such as autonomous systems, scientific experimentation, and multi-task planning.

Our contributions are primarily theoretical and do not involve the deployment of new systems or direct interaction with users or sensitive data. As such, we do not identify any specific negative societal consequences that require mitigation at this stage.

## Acknowledgments

This project is supported by the Israel Science Foundation (ISF, grant number 2250/22).

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

# A. Omitted details for Section 4

To prove the results in this section, we first define the "good event" $E_{EPC}$, the probabilistic event that:

1. The statements in Lemma A.3 hold.
2. For all significant $s, a, h$ w.r.t. $\pi_{\exp}$, $\sum_{t=1}^{T} \mathbb{E}_{t-1} \left[ \frac{1}{N_h^t(s,a)} \right] \leq \sum_{t=1}^{T} \left[ \frac{2}{N_h^t(s,a)} \right] + 8 \ln \frac{2}{\delta}$, where $\mathbb{E}_{t-1}$ signifies expectation conditioned on all randomness prior to episode $t$.

**Lemma A.1.** *The good event $E_{EPC}$ holds w.p. at least $1 - \delta$.*

*Proof.* By applying Lemma A.3 with confidence parameter $\delta/2$, and Lemma D.3 using confidence parameter $\delta/(2|S||A|H)$, and then taking a union bound over all $s, a, h$ and the two events, we obtain an overall confidence level of $1 - \delta$. □

## A.1. Online mirror descent

The occupancy measure update in each iteration is executed using OMD. Formally, the optimization problem to solve, in order to get the next occupancy measure, given $P$ is

$$\mu^{t+1} = \arg\max_{\mu \in \Lambda(P)} \{\eta \langle \mu, r^t \rangle - D_R(\mu \parallel \mu^t)\} \tag{8}$$

where

$$D_R(\mu || \mu') = \sum_{s,a,h,s'} \mu_h(s, a, s') \ln \frac{\mu_h(s, a, s')}{\mu_h'(s, a, s')} - \sum_{s,a,h,s'} \mu_h(s, a, s') + \sum_{s,a,h,s'} \mu_h'(s, a, s').$$

## A.2. Exploration Policy Algorithm

---
**Algorithm 5** UC-O-REPS
---
1: **input:** state space $S$, action space $A$, horizon $H$, number of episodes $T$, reward upper bound up to time $T$ $\mathcal{G}(T)$, $||r^t||_\infty$ bound $B$.
2: **init:** $N_h^1(s,a) \leftarrow 0$, $\mu_h^1(s, a, s') \leftarrow \frac{1}{|S|^2|A|}, \forall(s, a, s', h)$, $\pi_h^1(a|s) \leftarrow \frac{1}{|A|}, \ \forall(s, a, h)$, $\eta \leftarrow \sqrt{\frac{4H \log(|S||A|)}{B \cdot \mathcal{G}(T)}}$.
3: **for** $t = 1, ..., T$ **do**
4:     **traverse** trajectory $(s_1^t, a_1^t, s_2^t, a_2^t, ..., s_{H+1}^t)$ using policy $\pi^t$.
5:     $N_h^t(s, a, s') \leftarrow \sum_{n=1}^{N} \mathbb{I}[s_h^n = s, a_h^n = a, s_{h+1}^n = s']$.
6:     $N_h^t(s, a) \leftarrow \sum_{s' \in S} N_h^t(s, a, s')$.
7:     **calculate** $\hat{P}_h^t(s' \mid s, a) = \frac{N_h^t(s,a,s')}{\max\{1, N_h^t(s,a)\}}$.
8:     **observe** reward function $r_t$.
9:     **calculate** $\mu^{t+1}$ according to Eq. (13).
10:     **update** $\pi^{t+1}$ using Eq. (10).
11: **end for**
---

The chosen exploration policy algorithm for this work is described in Algorithm 5. This algorithm is a variation of Rosenberg & Mansour (2019). The procedure is identical in spirit with two main differences. First, we use Bernstein inequality to estimate the dynamics as in Jin et al. (2020a), yielding tighter regret guarantees. Unlike Jin et al. (2020a), however, our algorithm does not allow bandit feedback on the rewards, which once again allows for an improved regret bound. Second, we provide a first-order regret bound that depends on the cumulative reward of the learner.

We prove the following guarantee for this algorithm.

**Lemma A.2.** *Suppose that $E_{EPC}$ holds. Let $||r^t||_\infty \leq B$ for all $t$. Then, if we run Algorithm 5 for $T$ episodes, the following holds:*

$$\textbf{Reg}_{UC\text{-}O\text{-}REPS}(T) \leq 2\sqrt{BH \log(|S||A|)\mathcal{G}(T)} + \sqrt{360B|S|^2|A|H^2\ell_0^2 \cdot \mathcal{G}(T)} + 300B|S|^2|A|H^3\ell_0^2.$$

*where $\mathcal{G}(T)$ is an upper bound on the cumulative reward from all episodes: $\sum_{t=1}^{T} \langle r^t, \mu^t \rangle \leq \mathcal{G}(T)$.*

The proof of Lemma A.2 is deferred to Section A.3.3. Below we give a broad explanation of our algorithm.

In addition, we note that we decompose the regret of Algorithm 5 as

$$\mathbf{Reg}_{\text{UC-O-REPS}}(T) = \sum_{t=1}^{T}\langle r^t,\ \mu^{P,\pi^\star}\rangle - \sum_{t=1}^{T}\langle r^t,\ \mu^{P,\pi_t}\rangle \tag{9}$$

$$= \underbrace{\sum_{t=1}^{T}\left(\langle r^t,\ \mu^{\hat{P}^t,\pi_t}\rangle - \langle r^t,\ \mu^{P,\pi_t}\rangle\right)}_{:=\hat{R}_{1:T}^{Dynamics}} + \underbrace{\sum_{t=1}^{T}\left(\langle r^t,\ \mu^{P,\pi^\star}\rangle - \langle r^t,\ \mu^{\hat{P}^t,\pi_t}\rangle\right)}_{:=\hat{R}_{1:T}^{Policy}},$$

where $\mu^{P,\pi}$ is the occupancy measure induced by dynamics - $P$ and policy - $\pi$, $\hat{P}^t$ is the estimation of $P$ at episode $t$, $\pi^t$ is the policy estimation at episode $t$ and $\pi^\star$ is the optimal policy. Note that $\hat{R}_{1:T}^{Dynamics}$ represents the error that comes from the estimation of the unknown transition function, and $\hat{R}_{1:T}^{Policy}$ is related to the specific algorithm chosen for policy estimation (OMD in our case).

To account for the unknown dynamics, our algorithm operates over the occupancy measure space of an augmented MDP with state space $S$ and action space $A \times S$. The augmented action $(a, s')$ at state $s$ yields a reward $r(s, a)$ and transitions to state $s'$. Occupancy measures in this augmented MDP have the form $\mu : S \times A \times S \times [H]$ such that for policy $\pi$, $\mu_h(s, a, s') = \Pr_\pi[s_h = s, a_h = a, s_{h+1} = s']$. Moreover, Rosenberg & Mansour (2019) prove there is a one-to-one correspondence between such an occupancy measure $\mu$ and a pair of policy and dynamics, given by:

$$\pi_h^\mu(a \mid s) = \frac{\sum_{s'} \mu_h(s, a, s')}{\sum_{a',s'} \mu_h(s, a', s')}, \qquad P_h^\mu(s' \mid s, a) = \frac{\mu_h(s, a, s')}{\sum_{s''} \mu_h(s, a, s'')}. \tag{10}$$

The algorithm essentially runs mirror descent in occupancy measure space that is continuously updated with more samples from the transition model. Thus each prediction of an occupancy measure corresponds to a policy, which the algorithm executes on the true MDP, and transitions that are within a confidence interval around the true transitions.

The confidence set $\mathcal{P}^t$ at episode $t$ consists of all transition kernels whose entries are close to the empirical estimate:

$$\mathcal{P}^t := \{P : |P_h(s' \mid s, a) - \hat{P}_h^t(s' \mid s, a)| \le \epsilon_h^t(s' \mid s, a)\}, \tag{11}$$

where the confidence radius is defined as

$$\epsilon_h^t(s' \mid s, a) := 2\sqrt{\frac{\hat{P}_h^t(s'|s,a)\ell_0}{\max\{N_h^t(s,a),\ 1\}}} + \frac{14\ell_0}{3\max\{N_h^t(s,a),\ 1\}}, \tag{12}$$

Lemma A.3 provides the guarantee that $P \in \mathcal{P}^t$, i.e., the true transition kernel is contained in $\mathcal{P}^t$, with high probability $\forall t$.

The OMD updates in our algorithm are identical to the one done by (Jin et al., 2020a), and represent a convex optimization problem with linear constraints, which can be solved in polynomial time (e.g. using newton's method with dimensionality of $n = O(|S|^2|A|H)$ and $m = O(|S|^2|A|H)$ linear inequality constraints, we get about $O((m+n)^3) = O(|S|^6|A|^3H^3)$ time).

This optimization problem can be further reformulated into a dual problem, which is a convex optimization problem with only non-negativity constraints, and thus can be solved more efficiently. The full derivation can be found in Jin et al., 2020a, Appendix A.1. The constraints on the optimization using $\Lambda(\mathcal{P}^t)$ are given below.

**Lemma A.3** (Jin et al., 2020a, Lemma 2). *Let $\epsilon_h^t(s' \mid s, a)$ be defined as in Eq. (12), then w.p. at least $1 - \delta/2$, $P \in \mathcal{P}^t$ for all $t$.*

Lastly, our algorithm takes an online mirror descent step in the convex space defined by $\Lambda(\mathcal{P}^t)$ (defined below), that correspond to transition kernels from $\mathcal{P}_t$, as follows

$$\mu^{t+1} = \arg\max_{\mu \in \Lambda(\mathcal{P}^t)}\{\eta\langle \mu, r^t\rangle - D_R(\mu \,\|\, \mu^t)\}, \tag{13}$$

where $R$ is the negative entropy regularizer, defined as $R(\mu) = \sum_{s,a,h,s'} \mu_h(s, a, s') \log(\mu_h(s, a, s')) - \sum_{s,a,h,s'} \mu_h(s, a, s')$. Here, the set $\Lambda(\mathcal{P}^t)$ is the set of occupancy measures that satisfy the following linear equations (see

Jin et al., 2020a for further details):

$$
\begin{aligned}
\forall h \in [H]: & \quad \sum_{s,a,s'} \mu_h(s,a,s') = 1; \\
\forall h \in [H-1], s \in S: & \quad \sum_{s',a} \mu_h(s',a,s) = \sum_{a,s'} \mu_{h+1}(s,a,s'); \\
\forall s,a,s',h: & \quad \mu_h(s,a,s') \leq [\hat{P}_h^t(s' \mid s,a) + \epsilon_h^t(s' \mid s,a)] \cdot \sum_{s'} \mu_h(s,a,s'); \\
\forall s,a,s',h: & \quad \mu_h(s,a,s') \geq [\hat{P}_h^t(s' \mid s,a) - \epsilon_h^t(s' \mid s,a)] \cdot \sum_{s'} \mu_h(s,a,s'); \\
\forall s,a,s',h: & \quad \mu_h(s,a,s') \geq 0.
\end{aligned}
\tag{14}
$$

### A.3. Proof of lemmas

#### A.3.1. LEMMA 4.4 PROOF

*Proof.* We prove this in two parts, the first part is correct for all $s, a, h$, and in the second part we need to address the significant and non significant $s, a, h$ in a different manner.

For all $s, a, h$ the following derivation holds with probability at least $1 - \delta/4$:

$$
\begin{aligned}
\max_\mu \sum_{t=1}^T \langle \mu, \lambda_t \rangle = \sum_{t=1}^T \langle \mu^\star, \lambda^t \rangle &= \sum_{t=1}^T \sum_{s,a,h} \frac{\mu_h^\star(s,a)}{c + N_h^t(s,a)} \overset{(I)}{\geq} \sum_{t=1}^T \sum_{s,a,h} \frac{\mu_h^\pi(s,a)}{c + N_h^t(s,a)} \\
&\overset{(II)}{\geq} \sum_{s,a,h} \frac{\mu_h^\pi(s,a)}{c/T + N_h^T(s,a)/T} \overset{(III)}{\geq} \sum_{s,a,h} \frac{\mu_h^\pi(s,a)}{\frac{c}{T} + \frac{4}{T} \cdot \ln\left(\frac{2|S||A|H}{\delta}\right) + \frac{3}{2}\mu_h^T(s,a)} \\
&\overset{(IV)}{\geq} \sum_{s,a,h} \frac{\mu_h^\pi(s,a)}{\frac{2c}{T} + \frac{3}{2}\mu_h^T(s,a)} = \sum_{(s,a,h)\in\psi} \frac{\mu_h^\pi(s,a)}{\frac{2c}{T} + \frac{3}{2}\mu_h^T(s,a)} + \sum_{(s,a,h)\notin\psi} \frac{\mu_h^\pi(s,a)}{\frac{2c}{T} + \frac{3}{2}\mu_h^T(s,a)}
\end{aligned}
$$

where (I) is following the optimality of $\mu^\star$, (II) is true since $\lambda_t$ are monotonic descending with $t$ and (III) holds with probability at least $1 - \delta/4$ according to Lemma D.3 and (IV) since $c = 4|S|H^2\ell_0$. Now, in order to move forward, we need to separate the significant and non significant $s, a, h$.

$\forall \omega$-significant $s, a, h$ with respect to $\pi_{\text{exp}}$:

$$
\sum_{(s,a,h)\in\psi} \frac{\mu_h^\pi(s,a)}{\frac{2c}{T} + \frac{3}{2}\mu_h^T(s,a)} \geq \sum_{(s,a,h)\in\psi} \frac{2\mu_h^\pi(s,a)}{5\mu_h^T(s,a)}.
$$

This is following Definition 4.1 and the assumption that $\frac{2c}{T} \leq \omega$.

$\forall \omega$-non significant $s, a, h$ with respect to $\pi_{\text{exp}}$:

$$
\sum_{(s,a,h)\notin\psi} \frac{\mu_h^\pi(s,a)}{\frac{2c}{T} + \frac{3}{2}\mu_h^T(s,a)} \overset{(I)}{\geq} \sum_{(s,a,h)\notin\psi} \frac{\mu_h^\pi(s,a)}{\frac{2c}{T} + \frac{3}{2}\omega} \overset{(II)}{\geq} \sum_{(s,a,h)\notin\psi} \frac{2\mu_h^\pi(s,a)}{5\omega}
$$

where (I) is following Definition 4.1 and (II) is true according to assumption that $\frac{2c}{T} \leq \omega$. $\square$

#### A.3.2. LEMMA 4.5 PROOF

*Proof.*

$$
\begin{aligned}
\sum_{t=1}^T \langle \mu^t, \lambda^t \rangle &= \sum_{t=1}^T \sum_{s,a,h} \mu_h^t(s,a)\lambda_h^t(s,a) = \sum_{t=1}^T \sum_{s,a,h} \frac{\mu_h^t(s,a)}{c + N_h^t(s,a)} \\
&\overset{(I)}{\leq} \sum_{t=1}^T \sum_{s,a,h} \frac{\mu_h^t(s,a)}{4\ell_0 + N_h^t(s,a)} \overset{(II)}{\leq} \sum_{t=1}^T \sum_{s,a,h} \frac{2\mu_h^t(s,a)}{\sum_{j=1}^t \mu_h^j(s,a)} \overset{(III)}{\leq} 4\sum_{s,a,h} \ln T = 4|S||A|H\ln T
\end{aligned}
$$

where we have (I) since $c \geq 4\ell_0$, (II) holds with probability at least $1 - \delta/4$ according to Lemma D.3 and (III) following Lemma D.2. $\square$

### A.3.3. LEMMA A.2 PROOF

As described in Section A.2, the regret of the algorithm can be represented as $\hat{R}_{1:T}^{Dynamics} + \hat{R}_{1:T}^{Policy}$. In the following, we prove an upper bound for each of them.

**Lemma A.4** ($\hat{R}_{1:T}^{Dynamics}$ bound). *Suppose that $E_{EPC}$ holds. If we run Algorithm 5 for $T$ episodes, with loss function $||r^t||_\infty \leq B$, the following holds*

$$\hat{R}_{1:T}^{Dynamics} \leq \sqrt{360B|S|^2|A|H^2 \cdot \mathcal{G}(T) \cdot \ell_0^2} + 300B|S|^2|A|H^3 \cdot \ell_0^2.$$

*where $\mathcal{G}(T)$ is an upper bound on the cumulative reward.*

*Proof.* Using Lemma D.5 we have:

$$\hat{R}_{1:T}^{Dynamics} = \sum_{t=1}^{T} \left| V^{\pi_t}(s_1) - \hat{V}^{\pi_t}(s_1) \right|$$

$$\leq 3 \cdot \sum_{t=1}^{T} \sqrt{HB \cdot V^{\mathbb{P},\pi}(s_1)} \cdot \sqrt{\mathbb{E}_{t-1} \sum_{h=1}^{H} \frac{C_1|S|}{\max\{1, N_h^t(s,a)\}}} + \sum_{t=1}^{T} \mathbb{E}_{t-1} \sum_{h=1}^{H} \frac{6BC_1H^2|S|}{\max\{1, N_h^t(s,a)\}}$$

$$\leq 3 \cdot \sqrt{HB \cdot \sum_{t=1}^{T} V^{\mathbb{P},\pi}(s_1)} \cdot \sqrt{\sum_{h=1}^{H}\sum_{t=1}^{T} \mathbb{E}_{t-1} \frac{C_1|S|}{\max\{1, N_h^t(s,a)\}}} + \sum_{h=1}^{H}\sum_{t=1}^{T} \mathbb{E}_{t-1} \frac{6BC_1H^2|S|}{\max\{1, N_h^t(s,a)\}}$$

$$\leq 3 \cdot \sqrt{HB \cdot \sum_{t=1}^{T} V^{\mathbb{P},\pi}(s_1)} \cdot \sqrt{\sum_{h=1}^{H}\sum_{t=1}^{T} \left[ \frac{2C_1|S|}{\max\{1, N_h^t(s,a)\}} \right] + 8C_1|S|\ell_0} \qquad (E_{EPC})$$

$$+ \sum_{h=1}^{H}\sum_{t=1}^{T} \left[ \frac{12BC_1H^2|S|}{\max\{1, N_h^t(s,a)\}} \right] + 48 \cdot BC_1H^2|S|\ell_0$$

$$\leq 3 \cdot \sqrt{HB \cdot \sum_{t=1}^{T} V^{\mathbb{P},\pi}(s_1)} \cdot \sqrt{\sum_{s=1}^{|S|}\sum_{a=1}^{|A|}\sum_{h=1}^{H}\sum_{t=1}^{N_h^t(s,a)} \frac{2C_1|S|}{t} + 8C_1|S|\ell_0}$$

$$+ \sum_{s=1}^{|S|}\sum_{a=1}^{|A|}\sum_{h=1}^{H}\sum_{t=1}^{N_h^t(s,a)} \frac{12BC_1H^2|S|}{t} + 48 \cdot BC_1H^2|S|\ell_0$$

$$\leq 3 \cdot \sqrt{2BC_1|S|^2|A|H^2 \ln(T)\mathcal{G}(T)} + 3 \cdot \sqrt{8C_1BH|S|\ell_0\mathcal{G}(T)} \qquad (\text{definition of } \mathcal{G}(T))$$

$$+ 12BC_1|S|^2|A|H^3 \ln T + 48 \cdot BC_1H^2|S|\ell_0$$

$$\leq \sqrt{360B|S|^2|A|H^2 \cdot \mathcal{G}(T) \cdot \ell_0^2} + 300B|S|^2|A|H^3 \cdot \ell_0^2. \qquad (\text{placing } C_1 = 5\ell_0)$$

$\square$

**Lemma A.5** ($\hat{R}_{1:T}^{Policy}$ bound). *Suppose that $E_{EPC}$ holds. If we run Algorithm 5 for $T$ episodes, with loss function $||r^t||_\infty \leq B$, the following holds*

$$\hat{R}_{1:T}^{Policy} \leq 2\sqrt{BH\log(|S||A|)\mathcal{G}(T)},$$

*where $\mathcal{G}(T)$ is an upper bound on the cumulative reward from all episodes.*

*Proof.* Algorithm 5 uses online mirror descent (OMD) in order to find $\pi_{t+1}$ at each iteration (solving Eq. (13), and using Eq. (3)).

Now, using Rosenberg & Mansour (2019, Theorem 5.3) together with Lemma A.3, we obtain:

$$\sum_{t=1}^{T} D_R(\mu^t \| \mu^{t+1}) \le \frac{\eta^2}{2} \sum_{t=1}^{T} \sum_{s,a,h} \mu_h^t(s,a)(r_h^t(s,a))^2$$

$$\implies \sum_{t=1}^{T} D_R(\mu^t \| \mu^{t+1}) \le \frac{\eta^2}{2} \sum_{t=1}^{T} \sum_{s,a,h} \mu_h^t(s,a)(r_h^t(s,a))^2 \le \frac{B\eta^2}{2} \cdot \underbrace{\sum_{t=1}^{T} \sum_{s,a,h} \mu_h^t(s,a)r_h^t(s,a)}_{\le \mathcal{G}(T)}.$$

Similarly,

$$D_R(\mu \| \mu^1) = \sum_{s,a,h,s'} \mu_h(s,a,s') \log \frac{\mu_h(s,a,s')}{\mu_h^1(s,a,s')} - \sum_{s,a,h,s'} \mu_h(s,a,s') + \sum_{s,a,h,s'} \mu_h^1(s,a,s')$$

$$\le \sum_{s,a,h,s'} \mu_h(s,a,s') \log \frac{1}{\mu_h^1(s,a,s')} = \sum_{s,a,h,s'} \mu_h(s,a,s') \log(|S|^2|A|) \le 2H \log(|S||A|)$$

Combining the two bounds together in Theorem 2.1, we get the bound for $\hat{R}_{1:T}^{Policy}$:

$$\hat{R}_{1:T}^{Policy} \le \frac{\eta B}{2} \cdot \mathcal{G}(T) + \frac{2}{\eta} \cdot H \log(|S||A|).$$

Choosing $\eta = \sqrt{\frac{4H \log(|S||A|)}{B \cdot \mathcal{G}(T)}}$ yields $\hat{R}_{1:T}^{Policy} \le 2\sqrt{BH \log(|S||A|)\mathcal{G}(T)}$. $\qquad\square$

# B. Omitted details for Section 5

First, we define the following helpful quantity: Given any distribution $\rho \in \Delta(S)$ and any function $V : \mathcal{S} \mapsto \mathbb{R}$, we define the associated variance as

$$\text{Var}_\rho(V) := \sum_{s \in S} \rho(s)V(s)^2 - \left( \sum_{s \in S} \rho(s)V(s) \right)^2.$$

An effective strategy for offline reinforcement learning is to adopt the *pessimism in the face of uncertainty* principle. This approach requires a careful quantification of the uncertainty in value estimation. We now describe how this uncertainty is modeled in our algorithm. Specifically, for each $(s,a,h) \in \mathcal{S} \times \mathcal{A} \times [H]$, we introduce a penalty term $b_h(s,a)$ based on Bernstein-style concentration inequalities.

**Reward-agnostic setting.** In this case, the Bernstein-style penalty is defined as

$$b_h(s,a) = \min \left\{ \sqrt{\frac{c_a \ell_0}{N_h(s,a)} \text{Var}_{\hat{P}_h(\cdot|s,a)}(\hat{V}_{h+1})} + c_a H \frac{\ell_0}{N_h(s,a)}, \ H \right\} \tag{15}$$

**Reward-free setting.** In this case, the Bernstein-style penalty is defined as

$$b_h(s,a) = \min \left\{ \sqrt{\frac{c_f |S| \ell_0}{N_h(s,a)} \text{Var}_{\hat{P}_h(\cdot|s,a)}(\hat{V}_{h+1})} + c_f |S| H \frac{\ell_0}{N_h(s,a)}, \ H \right\} \tag{16}$$

where $c_a$, $c_f > 0$ are universal constants, to be defined later. We note that the dependency of Eq. (15) in the size of the reward class is logarithmic, and as such, it is absorbed to $\ell_0$. The intuition behind this penalty is that we design it to interact cleanly with the expressions from Lemmas B.9 and B.11. Now, we are presenting two Lemmas that are relevant for both reward-agnostic and reward-free settings.

To prove the result we first define the "good events:"

1. $E_{RAE}$ - the probabilistic event that:
   - The statements in Theorem 4.2 hold.
   - The statements in Lemma B.9 hold.
   - The statements in Lemma B.3 hold.
   - For all significant $s, a, h$ w.r.t. $\pi_{\exp}$, $N_h(s, a) \geq \frac{N}{16} \cdot \mu_h^T(s, a)$.

2. $E_{RFE}$ - the probabilistic event that:
   - The statements in Theorem 4.2 hold.
   - The statements in Lemma B.11 hold.
   - The statements in Lemma B.3 hold.
   - For all significant $s, a, h$ w.r.t. $\pi_{\exp}$, $N_h(s, a) \geq \frac{N}{16} \cdot \mu_h^T(s, a)$.

**Lemma B.1.** *The good event $E_{RAE}$ holds w.p. at least $1 - \delta$.*

*Proof.* By applying Theorem 4.2, Lemma B.9 with confidence parameter $\delta/4$, and Lemmas B.3 and B.4 using confidence parameter $\delta/(4|S||A|H)$, and then taking a union bound over all $s, a, h$ and the four events, we obtain an overall confidence level of $1 - \delta$. $\qquad\square$

**Lemma B.2.** *The good event $E_{RFE}$ holds w.p. at least $1 - \delta$.*

*Proof.* By applying Theorem 4.2, Lemma B.11 with confidence parameter $\delta/4$, and Lemmas B.3 and B.4 using confidence parameter $\delta/(4|S||A|H)$, and then taking a union bound over all $s, a, h$ and the four events, we obtain an overall confidence level of $1 - \delta$. $\qquad\square$

Next, we present two useful lemmas that pertain to the *two-fold-subsampling* trick (Li et al., 2024a, Algorithm 3), used in Algorithm 3.

**Lemma B.3** (Li et al., 2024a, Lemma 3). *Suppose that the $N$ trajectories in $\mathcal{D}$ are generated in an i.i.d. fashion. With probability at least $1 - \delta$, $N_h^{trim}(s)$ and $N_h^{trim}(s, a)$ obey:*

$$N_h^{trim}(s) \quad \leq N_h^{main}(s),$$
$$N_h^{trim}(s, a) \geq \frac{N\mu_h(s, a)}{8} - 5\sqrt{N\mu_h(s, a)\log\frac{8NH}{\delta}},$$

*simultaneously for all $1 \leq h \leq H$ and all $(s, a) \in \mathcal{S} \times \mathcal{A}$.*

**Lemma B.4.** *Let $N_h^{trim}(s, a)$ be defined as in Li et al. (2024a, Algorithm 3), $N$ be the number of trajectories collected by Algorithm 3 and $\mu_h^T(s, a)$ be the occupancy measure associated with $\pi_{exp}$. Then, for all significant $s, a, h$, conditioned on $E_{RAE}/E_{RFE}$,*

$$N_h^{trim}(s, a) \geq \frac{N\mu_h^T(s, a)}{16}$$

*Proof.* We note that for

$$\mu_h^T(s, a) \geq \frac{80^2 \log(8NH\delta^{-1})}{N} \tag{17}$$

we have

$$\frac{N\mu_h^T(s, a)}{8} - 5\sqrt{N\mu_h^T(s, a)\log\frac{8NH}{\delta}} \geq \frac{N\mu_h^T(s, a)}{16}.$$

Now, for all significant $s, a, h$ Eq. (17) holds since

$$\mu_h^T(s, a) \underbrace{\geq}_{Definition\ 4.1} \omega = \frac{\epsilon}{885|S||A|H^2\ell_0^2} \underbrace{\geq}_{for\ \epsilon < 1} \frac{80^2 \log(8NH\delta^{-1})}{N}$$

(Noting that $N = O(|S||A|H^3\ell_0^3/\epsilon^2)$ for RAE setting and $N = O(|S|^2|A|H^3\ell_0^3/\epsilon^2)$ for RFE setting). We finish the proof by using Lemma B.3. $\qquad\square$

**Lemma B.5.** *Suppose that $E_{RAE}/E_{RFE}$ holds (depending on the setting of $b_h(s,a)$). Let $\hat{Q}_h(s,a)$ and $\hat{V}_h(s)$ be defined as in Algorithm 4, then for all $(s,a,h)$ we have:*

$$\hat{Q}_h(s,a) \le Q_h^{\hat{\pi}}(s,a) \quad \text{and} \quad \hat{V}_h(s) \le V_h^{\hat{\pi}}(s).$$

*Proof.* The claim from the lemma holds trivially for the base case with $h = H+1$, given that $\hat{Q}_{H+1}(s,a) = Q_{H+1}^{\hat{\pi}}(s,a) = 0$. Next, suppose that $\hat{Q}_{h+1}(s,a) \le Q_{h+1}^{\hat{\pi}}(s,a)$ holds for all $(s,a) \in \mathcal{S} \times \mathcal{A}$ and some step $h+1$. We would like to show that the claimed inequality holds for step $h$ as well. First, we show that the second claim of the lemma is trivial given the first one.

$$\hat{V}_h(s) = \max_a \hat{Q}_h(s,a) = \hat{Q}_h(s,\hat{\pi}_h(s)) \le Q_h^{\hat{\pi}}(s,\hat{\pi}_h(s)) = V_h^{\hat{\pi}}(s). \tag{18}$$

Secondly, if $\hat{Q}_h(s,a) = 0$ the claim holds trivially, and otherwise our update rule reveals that:

$$\begin{aligned}
\hat{Q}_h(s,a) &= r_h(s,a) + \hat{P}_{h,s,a}\hat{V}_{h+1} - b_h(s,a) \\
&= r_h(s,a) + P_{h,s,a}\hat{V}_{h+1} + \left(\hat{P}_{h,s,a} - P_{h,s,a}\right)\hat{V}_{h+1} - b_h(s,a) \\
&\le r_h(s,a) + P_{h,s,a}V_{h+1}^{\hat{\pi}} \qquad\qquad \text{(Eq. (18) and Eq. (19)/Eq. (20))} \\
&= Q_h^{\hat{\pi}}(s,a).
\end{aligned}$$

We have thus proved the lemma via a standard induction argument. $\qquad\square$

**Lemma B.6.** *Suppose that $E_{RAE}/E_{RFE}$ holds (depending on the setting of $b_h(s,a)$). Let $\hat{V}_h(s)$ be defined as in Algorithm 4, $V_h^\star(s)$ and $V_h^{\hat{\pi}}(s)$ be defined as the value functions using $\pi^\star$, $\hat{\pi}$ respectively. Then*

$$V_h^\star(s) - \hat{V}_h(s) \le P_{h,s,\pi_h^\star(s)}\left(V_{h+1}^\star - \hat{V}_{h+1}\right) + 2b_h(s,\pi_h^\star(s)).$$

*Proof.* First, we notice that according to Lemma B.5 we have that

$$0 \le V_h^\star(s) - V_h^{\hat{\pi}}(s) \le V_h^\star(s) - \hat{V}_h(s) = Q_h^\star(s,\pi_h^\star(s)) - \max_a \hat{Q}_h(s,a) \le Q_h^\star(s,\pi_h^\star(s)) - \hat{Q}_h(s,\pi_h^\star(s)).$$

In addition, note that

$$Q_h^\star(s,\pi_h^\star(s)) = r(s,\pi_h^\star(s)) + P_{h,s,\pi_h^\star(s)}V_{h+1}^\star, \hat{Q}_h(s,\pi_h^\star(s)) = \max\left\{r(s,\pi_h^\star(s)) + \hat{P}_{h,s,\pi_h^\star(s)}\hat{V}_{h+1} - b_h(s,\pi_h^\star(s)), 0\right\}.$$

All together we have that

$$\begin{aligned}
V_h^\star(s) - \hat{V}_h(s) &\le r(s,\pi_h^\star(s)) + P_{h,s,\pi_h^\star(s)}V_{h+1}^\star - \left\{r(s,\pi_h^\star(s)) + \hat{P}_{h,s,\pi_h^\star(s)}\hat{V}_{h+1} - b_h(s,\pi_h^\star(s))\right\} \\
&= P_{h,s,\pi_h^\star(s)}V_{h+1}^\star - \hat{P}_{h,s,\pi_h^\star(s)}\hat{V}_{h+1} + b_h(s,\pi_h^\star(s)) \\
&= P_{h,s,\pi_h^\star(s)}\left(V_{h+1}^\star - \hat{V}_{h+1}\right) - \left(\hat{P}_{h,s,\pi_h^\star(s)} - P_{h,s,\pi_h^\star(s)}\right)\hat{V}_{h+1} + b_h(s,\pi_h^\star(s)) \\
&\le P_{h,s,\pi_h^\star(s)}\left(V_{h+1}^\star - \hat{V}_{h+1}\right) + 2b_h(s,\pi_h^\star(s)). \qquad \text{(Eq. (19)/Eq. (20))}
\end{aligned}$$

$\qquad\square$

**Lemma B.7.** *Let $\pi$ be a policy with respect to MDP with transition dynamics $P_h(\cdot \mid s,a)$ and the reward function $r_h(s,a)$. If $\|r\|_\infty \le B$ for $B > 0$, then*

$$\mathbb{E}_\pi\left[\sum_{h=1}^H \mathrm{Var}_{P_h(\cdot|s_h,a_h)}(V_{h+1}^\pi)\right] \le BHV_1^\pi(s_1).$$

*Proof.*

$$\mathbb{E}_\pi \left[ \sum_{h=1}^H \mathrm{Var}_{P_h(\cdot|s_h,a_h)}(V_{h+1}^\pi) \right]$$

$$= \mathbb{E}_\pi \left\{ \left( \sum_{h=1}^H \mathbb{E}_{s_{h+1}\sim P_h(\cdot|s_h,a_h)} \left[ V_{h+1}^\pi(s_{h+1}) - \mathbb{E}_{\hat{s}\sim P_h(\cdot|s_h,a_h)} V_{h+1}^\pi(\hat{s}) \right] \right)^2 \right\}$$

$$= \mathbb{E}_\pi \left\{ \left( \sum_{h=1}^H \mathbb{E}_{s_{h+1}\sim P_h(\cdot|s_h,a_h)} \left[ V_{h+1}^\pi(s_{h+1}) - \mathbb{E}_{\hat{s}\sim P_h(\cdot|s_h,a_h)} V_{h+1}^\pi(\hat{s}) + r_h(s_h,a_h) - r_h(s_h,a_h) \right] \right)^2 \right\}$$

$$= \mathbb{E}_\pi \left\{ \left( \sum_{h=1}^H \mathbb{E}_{s_{h+1}\sim P_h(\cdot|s_h,a_h)} \left[ V_{h+1}^\pi(s_{h+1}) - V_h^\pi(s_h) + r_h(s_h,a_h) \right] \right)^2 \right\} \qquad \text{(Bellman equations)}$$

$$= \mathbb{E}_\pi \left\{ \left( \sum_{h=1}^H \left[ r_h(s_h,a_h) - V_1^\pi(s_1) \right] \right)^2 \right\} \leq \mathbb{E}_\pi \left\{ \left( \sum_{h=1}^H r_h(s_h,a_h) \right)^2 \right\}$$

$$\leq BH\mathbb{E}_\pi \left\{ \sum_{h=1}^H r_h(s_h,a_h) \right\} = BHV_1^\pi(s_1)$$

$\square$

**Lemma B.8.** *Suppose that $E_{RAE}/E_{RFE}$ holds (depending on the setting of $b_h(s,a)$). Let $P$ be the transition dynamics of some MDP, $\hat{V}$ be defined as in Algorithm 4 and $V^\star$ be the value function of the optimal policy $\pi^\star$. If $\|V\|_\infty \leq H$, then for $Y \in \mathbb{R}$, we have that*

$$\sqrt{\frac{Y \cdot \mathrm{Var}_{P_h(\cdot|s,\pi_h^\star(s))}(\hat{V}_{h+1} - V_{h+1}^\star)}{N_h(s,a)}} \leq \frac{YH^2}{2N_h(s,a)} + \frac{1}{H} \sum_{s'} P_h(s' \mid s, \pi_h^\star(s)) \left( V_{h+1}^\star(s') - \hat{V}_{h+1}(s') \right)$$

*Proof.*

$$\sqrt{\frac{Y \cdot \mathrm{Var}_{P_h(\cdot|s,\pi_h^\star(s))}(\hat{V}_{h+1} - V_{h+1}^\star)}{N_h(s,a)}}$$

$$\leq \sqrt{\frac{Y \cdot \sum_{s'} P_h(s' \mid s, \pi_h^\star(s)) \left( \hat{V}_{h+1}(s') - V_{h+1}^\star(s') \right)^2}{N_h(s,a)}} \qquad \text{(variance definition)}$$

$$\leq \frac{YH^2}{2N_h(s,a)} + \frac{1}{H^2} \sum_{s'} P_h(s' \mid s, \pi_h^\star(s)) \left( \hat{V}_{h+1}(s') - V_{h+1}^\star(s') \right)^2 \qquad \text{(AM-GM)}$$

$$\leq \frac{YH^2}{2N_h(s,a)} + \frac{1}{H} \sum_{s'} P_h(s' \mid s, \pi_h^\star(s)) \left( V_{h+1}^\star(s') - \hat{V}_{h+1}(s') \right) \qquad (\|V\|_\infty \leq H, \text{Lemma B.5})$$

$\square$

## B.1. reward-agnostic offline RL

**Lemma B.9** (Li et al., 2024a, Lemma 8). *Consider any $1 \leq h \leq H$, and any vector $V \in \mathbb{R}^S$ independent of $\hat{P}_h$ obeying $\|V\|_\infty \leq H$. With probability at least $1 - 4\delta/H$, one has*

$$\left| \left( \hat{P}_{s,a,h} - P_{s,a,h} \right) V \right| \leq \sqrt{\frac{48 \, \mathrm{Var}_{\hat{P}_{s,a,h}}(V)\ell_0}{N_h(s,a)}} + \frac{48H\ell_0}{N_h(s,a)}$$

$$\mathrm{Var}_{\hat{P}_{h,s,a}}(V) \leq 2 \, \mathrm{Var}_{P_{s,a,h}}(V) + \frac{5H^2\ell_0}{3N_h(s,a)}$$

*simultaneously for all* $(s, a) \in \mathcal{S} \times \mathcal{A}$.

Note that given that Algorithm 4 works backwards, the iterate $\hat{V}_{h+1}$ does not use $\hat{P}_h$, and is hence statistically independent from $\hat{P}_h$. Thus, we can apply Lemma B.9 to obtain

$$\left| \left( \hat{P}_{h,s,a} - P_{h,s,a} \right) \hat{V}_{h+1} \right| \leq \sqrt{\frac{48 \operatorname{Var}_{\hat{P}_{s,a,h}}(\hat{V}_{h+1}) \ell_0}{N_h(s,a)}} + \frac{48 H \ell_0}{N_h(s,a)} \leq b_h(s,a) \tag{19}$$

in the presence of the Bernstein-style penalty Eq. (15), provided that the constant $c_a \geq 48$.

**Lemma B.10.** *Suppose that $E_{RAE}$ holds. Let $V_h^{\pi}(s_1)$ be the value function under dynamics $\mathbb{P}$ and policy $\pi$, and let $\|r\|_{\infty} \leq 1$ ($\Rightarrow \|V\|_{\infty} \leq H$) to be the reward function. Define $N(s_h, a_h)$ to be the number of samples given for $(s_h, a_h)$, and $H$ to be the horizon of the settings. In addition, let the penalty term be defined as in Eq. (15). In that case,*

$$\left| V_1^{\star}(s) - V_1^{\hat{\pi}}(s) \right| \leq 3 \cdot \sqrt{\mathbb{E} \sum_{h=1}^{H} \frac{16 H^2 c_a \ell_0}{N_h(s,a)} + \mathbb{E} \sum_{h=1}^{H} \frac{48 c_a H^2 \ell_0}{N_h(s,a)}}$$

*simultaneously for all* $(s, a) \in \mathcal{S} \times \mathcal{A}$.

*Proof.* Using the above derivations, we can now bound the error term for time $h$

$$V_h^{\star}(s) - V_h^{\hat{\pi}}(s) \leq P_h(\cdot \mid s, \pi_h^{\star}(s)) \left( V_{h+1}^{\star} - \hat{V}_{h+1} \right)(s) + 2 b_h(s, \pi_h^{\star}(s)) \qquad \text{(Lemmas B.5 and B.6)}$$

$$= P_h(\cdot \mid s, \pi_h^{\star}(s)) \left( V_{h+1}^{\star} - \hat{V}_{h+1} \right)(s) + \sqrt{\frac{4 c_a \ell_0 \operatorname{Var}_{\hat{P}_h(\cdot \mid s, \pi_h^{\star}(s))}(\hat{V}_{h+1})}{N_h(s,a)}} + \frac{2 c_a H \ell_0}{N_h(s,a)}$$

$$\leq P_h(\cdot \mid s, \pi_h^{\star}(s)) \left( V_{h+1}^{\star} - \hat{V}_{h+1} \right)(s) + \sqrt{\frac{8 c_a \ell_0 \operatorname{Var}_{P_h(\cdot \mid s, \pi_h^{\star}(s))}(\hat{V}_{h+1})}{N_h(s,a)}} + \frac{4 c_a H \ell_0}{N_h(s,a)} \qquad \text{(Lemma B.9)}$$

$$\leq P_h(\cdot \mid s, \pi_h^{\star}(s)) \left( V_{h+1}^{\star} - \hat{V}_{h+1} \right)(s) + \sqrt{\frac{16 c_a \ell_0 \operatorname{Var}_{P_h(\cdot \mid s, \pi_h^{\star}(s))}(\hat{V}_{h+1} - V_{h+1}^{\star})}{N_h(s,a)}}$$

$$+ \sqrt{\frac{16 c_a \ell_0 \operatorname{Var}_{P_h(\cdot \mid s, \pi_h^{\star}(s))}(V_{h+1}^{\star})}{N_h(s,a)}} + \frac{4 c_a H \ell_0}{N_h(s,a)}$$

$$\leq \left( 1 + \frac{1}{H} \right) \cdot P_h(\cdot \mid s, \pi_h^{\star}(s)) \left( V_{h+1}^{\star} - \hat{V}_{h+1} \right)(s) + \sqrt{\frac{16 c_a \ell_0 \operatorname{Var}_{P_h(\cdot \mid s, \pi_h^{\star}(s))}(V_{h+1}^{\star})}{N_h(s,a)}} + \frac{16 c_a H^2 \ell_0}{N_h(s,a)}, \qquad \text{(Lemma B.8)}$$

and unrolling the recursive equation to $h = 1$ yields

$$\left| V_1^{\star}(s) - \hat{V}_1(s) \right| \leq \mathbb{E} \left\{ \sum_{h=1}^{H} \left( 1 + \frac{1}{H} \right)^{H-h} \left( \sqrt{\frac{16 c_a \ell_0 \operatorname{Var}_{P_h(\cdot \mid s, \pi_h^{\star}(s))}(V_{h+1}^{\star})}{N_h(s,a)}} + \frac{16 c_a H^2 \ell_0}{N_h(s,a)} \right) \right\}$$

$$\leq 3 \sqrt{\mathbb{E} \sum_{h=1}^{H} \operatorname{Var}_{P_h(\cdot \mid s, \pi_h^{\star}(s))}(V_{h+1}^{\star})} \cdot \sqrt{\mathbb{E} \sum_{h=1}^{H} \frac{16 c_a \ell_0}{N_h(s,a)} + \mathbb{E} \sum_{h=1}^{H} \frac{48 c_a H^2 \ell_0}{N_h(s,a)}}$$

$$\leq 3 \cdot \sqrt{\mathbb{E} \sum_{h=1}^{H} \frac{16 H V_1^{\star}(s_1) c_a \ell_0}{N_h(s,a)} + \mathbb{E} \sum_{h=1}^{H} \frac{48 c_a H^2 \ell_0}{N_h(s,a)}}. \qquad \text{(Lemma B.7)}$$

$\square$

## B.2. Theorem 5.1 proof

*Proof.* Suppose that $E_{RAE}$ holds. Following Definition 4.1 for significance (with respect to $\pi_{\exp}$), we can perform the following decomposition:

$$\left|V_1^\star(s) - V_1^{\hat{\pi}}(s)\right| = \sum_{h=1}^H \mathbb{E}_{(s_h,a_h)\sim\mu_h^\pi} \left|V_h^{\pi^\star}(s_h) - Q_h^{\pi^\star}(s_h,a_h)\right| = \sum_{s,a,h} \left|V_h^{\pi^\star}(s_h) - Q_h^{\pi^\star}(s_h,a_h)\right| \mu_h^\pi(s,a)$$

$$= \underbrace{\sum_{(s,a,h)\in\Psi} \left|V_h^{\pi^\star}(s_h) - Q_h^{\pi^\star}(s_h,a_h)\right| \mu_h^\pi(s,a)}_{(I)} + \underbrace{\sum_{(s,a,h)\notin\Psi} \left|V_h^{\pi^\star}(s_h) - Q_h^{\pi^\star}(s_h,a_h)\right| \mu_h^\pi(s,a)}_{(II)}$$

$(I)$ For significant $(s,a,h)$, using Lemma B.10 we have:

$$\left|V_1^\star(s) - V_1^{\hat{\pi}}(s)\right| \le 3 \cdot \sqrt{\mathbb{E}\sum_{h=1}^H \frac{16H^2 c_a \ell_0}{N_h(s,a)} + \mathbb{E}\sum_{h=1}^H \frac{48 c_a H^2 \ell_0}{N_h(s,a)}}$$

$$\le 3 \cdot \sqrt{\sum_{(s,a,h)\in\Psi} \frac{256 c_a H^2 \ell_0 \cdot \mu_h^\pi(s,a)}{N \cdot \mu_h^T(s,a)} + \sum_{(s,a,h)\in\Psi} \frac{768 c_a H^2 \ell_0}{N \cdot \mu_h^T(s,a)} \cdot \mu_h^\pi(s,a)} \qquad (E_{RAE})$$

$$\le 3 \cdot \sqrt{\frac{75520 c_a |S||A|H^3 \ell_0^3}{N} + \frac{226560 c_a |S||A|H^3 \ell_0^3}{N}}. \qquad (\text{Theorem } 4.2)$$

$(II)$ For non-significant $(s,a,h)$, using Theorem 4.2 we have:

$$\sum_{(s,a,h)\notin\Psi} \left|V_h^{\pi^\star}(s_h) - Q_h^{\pi^\star}(s_h,a_h)\right| \mu_h^\pi(s,a) \le 2H \sum_{(s,a,h)\notin\psi} \mu_h^\pi(s,a) \le 295\omega|S||A|H^2\ell_0^2.$$

That means that all together we have:

$$\left|V_1^\star(s) - V_1^{\hat{\pi}}(s)\right| \le 3 \cdot \sqrt{\frac{75520 c_a |S||A|H^3 \ell_0^3}{N} + \frac{226560 c_a |S||A|H^3 \ell_0^3}{N}} + 295\omega|S||A|H^2\ell_0^2 \le \epsilon$$

For the choice of $N = \frac{81 \cdot 75520 c_a |S||A|H^3 \ell_0^3}{\epsilon^2}$ and $\omega = \frac{\epsilon}{885|S||A|H^2\ell_0^2}$ the lemma holds. $\qquad\square$

## B.3. reward-free offline RL

**Lemma B.11** (Li et al., 2024b, Lemma 5). *Let $\hat{P} := \{\hat{P}_h\}_{h=1}^H$ denote the empirical transition kernel constructed in Algorithm 3. With probability at least $1 - \delta$,*

$$\left|\left(\hat{P}_{s,a,h} - P_{s,a,h}\right)V\right| \le \sqrt{\frac{48|S|\operatorname{Var}_{\hat{P}_{s,a,h}}(V)\ell_0}{N_h(s,a)}} + \frac{64|S|H\ell_0}{N_h(s,a)}$$

$$\operatorname{Var}_{\hat{P}_{h,s,a}}(V) \le 8\operatorname{Var}_{P_{s,a,h}}(V) + \frac{10|S|H^2\ell_0}{N_h(s,a)}$$

*hold simultaneously for all $V \in \mathbb{R}^S$ obeying $\|V\|_\infty \le H$ and all $(s,a,h) \in S \times A \times [H]$.*

Similarly to reward-agnostic analysis, given that Algorithm 4 works backwards, the iterate $\hat{V}_{h+1}$ does not use $\hat{P}_h$, and is hence statistically independent from $\hat{P}_h$. Thus, we can apply Lemma B.11 to obtain

$$\left|\left(\hat{P}_{h,s,a} - P_{h,s,a}\right)\hat{V}_{h+1}\right| \le \sqrt{\frac{48|S|\operatorname{Var}_{\hat{P}_{s,a,h}}(\hat{V}_{h+1})\ell_0}{N_h(s,a)}} + \frac{64|S|H\ell_0}{N_h(s,a)} \qquad (20)$$

in the presence of the Bernstein-style penalty Eq. (15), provided that the constant $c_f \ge 64$.

**Lemma B.12.** *Suppose that $E_{RFE}$ holds. Let $V_h^\pi(s_1)$ be the value function under dynamics $\mathbb{P}$ and policy $\pi$, and let $||r||_\infty \leq 1 \ (\Rightarrow ||V||_\infty \leq H)$ to be the reward function. Define $N(s_h, a_h)$ to be the number of samples given for $(s_h, a_h)$, and $H$ to be the horizon of the settings. In addition, let the penalty term be defined as in Eq. (16). In that case,*

$$\left|V_1^\star(s) - V_1^{\hat{\pi}}(s)\right| \leq 3 \cdot \sqrt{\mathbb{E}\sum_{h=1}^{H} \frac{48|S|H^2 c_f \ell_0}{N_h(s,a)} + \mathbb{E}\sum_{h=1}^{H} \frac{144 c_f |S|H^2 \ell_0}{N_h(s,a)}}$$

*hold simultaneously for all $V \in \mathbb{R}^S$ obeying $||V||_\infty \leq H$ and all $(s, a, h) \in S \times A \times [H]$.*

The outline of the proof for Lemma B.12 is identical to the proof of Lemma B.10, where using Lemma B.11 instead of Lemma B.9, and defining $b_h(s, a)$ using Eq. (16) instead of Eq. (15).

### B.4. Theorem 5.2 proof

*Proof.* Suppose that $E_{RFE}$ holds. Following the proof of Theorem 5.1, we can see that the only change in the proof is regarding the significant $(s, a, h)$ part, which is given after the modification here using Lemma B.12, instead of Lemma B.10

$$\left|V_1^\star(s) - V_1^{\hat{\pi}}(s)\right| \leq 3 \cdot \sqrt{\mathbb{E}\sum_{h=1}^{H} \frac{48|S|H^2 c_f \ell_0}{N_h(s,a)} + \mathbb{E}\sum_{h=1}^{H} \frac{144 c_f |S|H^2 \ell_0}{N_h(s,a)}}$$

$$\leq 3 \cdot \sqrt{\sum_{(s,a,h)\in\Psi} \frac{768 c_f |S|H^2 \ell_0 \cdot \mu_h^\pi(s,a)}{N \cdot \mu_h^T(s,a)} + \sum_{(s,a,h)\in\Psi} \frac{2304 c_f |S|H^2 \ell_0}{N \cdot \mu_h^T(s,a)} \cdot \mu_h^\pi(s,a)} \qquad (E_{RFE})$$

$$\leq 3 \cdot \sqrt{\frac{226560 c_f |S|^2 |A| H^3 \ell_0^3}{N} + \frac{679680 c_f |S|^2 |A| H^3 \ell_0^3}{N}} \qquad \text{(Theorem 4.2)}$$

That means that all together we have:

$$\left|V_1^\star(s) - V_1^{\hat{\pi}}(s)\right| \leq 3 \cdot \sqrt{\frac{226560 c_f |S|^2 |A| H^3 \ell_0^3}{N} + \frac{679680 c_f |S|^2 |A| H^3 \ell_0^3}{N}} + 295\omega|S||A|H^2 \ell_0^2 \leq \epsilon$$

For the choice of $N = \frac{81 \cdot 226560 c_f |S|^2 |A| H^3 \ell_0^3}{\epsilon^2}$ and $\omega = \frac{\epsilon}{885|S||A|H^2 \ell_0^2}$ the lemma holds. $\qquad \square$

## C. Reward-free exploration lower bound

### C.1. Preliminaries

We start by formalizing the necessary components for the analysis. Define $\mathcal{E} = (S, A, H)$ to be an environment, containing the states in $S$, the actions in $A$ and horizon $H$. For a fixed environment, the transition class $\mathcal{P}$, is a class of transition and initial state distributions. A reward class $\mathcal{R}$, is a set of reward functions $r : (s, a, h) \to [0, 1]$. Given $\mathbb{P} \in \mathcal{P}$ and $r \in \mathcal{R}$, we denote $\mathcal{M}(\mathbb{P}, r)$ the MDP induced by $\mathbb{P}$ and $r$. In addition, let $l_{end} = \log_2 n$ be the depth of an $n$ leafed binary tree, and define $n$ as the largest power of 2 s.t. $4n \leq |S|$. In that case we also have that $n = \Omega(|S|)$.

### C.2. Single state

The single state construction and analysis are taken directly from Jin et al. (2020b), using the following lemma:

**Lemma C.1** (Lemma D.2 in Jin et al., 2020b). *Fix $\epsilon \leq 1$, $p \leq 1/2$, $A \geq 2$, and suppose that $n \geq c_0 \log_2 A$ for universal constants $c_0$. Then, there exists a distribution $\mathcal{D}$ over transition vectors $\mathbb{P} \in \mathcal{P}_{single}(\epsilon; n, A)$ such that any algorithm which $(\epsilon/12, p)$-learns the class $\mathcal{M}(\epsilon; n, A)$ satisfies*

$$\mathbb{E}_{\mathbb{P}\sim\mathcal{D}}\mathbb{E}_{\mathbb{P}, Alg}[K] \geq \frac{nA}{\epsilon^2}.$$

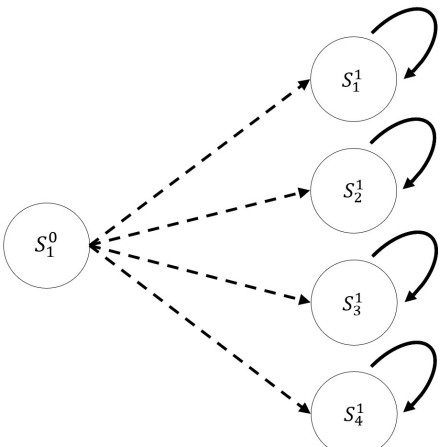

*Figure 2.* Single state lower bound scheme MDP construction for lower bound. Solid lines represent deterministic transition, and dashed lines represent probabilistic transitions.

Lemma C.1 shows a bound on the number of trajectories needed to learn the single state scheme, shown in Fig. 2. We now slightly extend the definition of single state transition class, to be conceptually time dependent, by defining $[\mathbb{P}_1^{(s_1^0)}, \ldots, \mathbb{P}_h^{(s_1^0)}]$ to be a vector of transition probabilities from state $s_1^0$, s.t. $\forall j \leq h,\ \mathbb{P}_j^{(x)} \in \mathcal{P}_{single}(\epsilon; n, A)$.

In addition, note that the transition probabilities from the initial state, to each of the $2n$ leafs, are close to uniform:

$$\left| \mathbb{P}_j^{(s_1^0)} \left[ s_x^1 \mid s_1^0,\ a \right] - \frac{1}{2n} \right| \leq \frac{\epsilon}{2n}$$

### C.3. Multiple state

To embed the single-state construction into a larger MDP, we construct a binary tree of depth $\log_2 n$, whose transitions are fully deterministic and known to the agent. Each leaf of this tree corresponds to an instance of the original single-state construction (total of $n$ instances). Moreover, the last layer of the MDP consists of $2n$ states, and every leaf of the binary tree is connected to all of the states in the final layer. Thus, ensuring that the overall structure preserves the properties of the single-state construction while situating it within a more complex MDP. See Fig. 1 for $n = 2$ example.

**Definition C.2** (state space). The number of states is $4n - 1$, where $n$ is assumed to be a power of 2. The first $2n - 1$ states are arranged in a binary tree configuration (with $n$ leafs), and the remaining $2n$ states are the last layer and are fully connected to all of the $n$ leafs of the binary tree. Define $s_i^l$ to be the $i$'th state at the $l$'th layer and $s_h$ to be the state at time $h$.

**Definition C.3** (deterministic actions). For the deterministic section of the MDP (up to layer $l_{end}$), define the following actions classes:

$$A^{(s)} = \left\{ \text{all}\ a\ \text{s.t.}\ \mathbb{P}_h \left[ s_1^0 \quad \mid s_1^0, a \right] = 1 \right\};$$

$$A^{(u)} = \left\{ \text{all}\ a\ \text{s.t.}\ \mathbb{P}_h \left[ s_{2x-1}^{l+1} \mid s_x^l, a \right] = 1 \right\};$$

$$A^{(d)} = \left\{ \text{all}\ a\ \text{s.t.}\ \mathbb{P}_h \left[ s_{2x}^{l+1} \quad \mid s_x^l, a \right] = 1 \right\},$$

where $A^{(s)}$ denotes the set of actions that deterministically keep the agent in the same state (green in Fig. 1), $A^{(u)}$ denotes the set of actions that deterministically transition the agent *upwards* (blue in Fig. 1), and $A^{(d)}$ denotes the set of actions that deterministically transition the agent *downwards* (red in Fig. 1). These action sets are forming a partition of the action space, that is $A = A^{(s)} \sqcup A^{(u)} \sqcup A^{(d)}$. For notational convenience, we use $a^{(s)}$, $a^{(u)}$, and $a^{(d)}$ to denote an arbitrary action in $A^{(s)}$, $A^{(u)}$, and $A^{(d)}$, respectively. In addition, observe that for any state $s_h \neq s_1^0$, the set $A^{(s)}$ is empty. Finally, we define $A_h^{(\star)} \subset A$ as the set of actions at time $h$ leading to the correct choice of $a^{(s)}$, $a^{(u)}$, $a^{(d)}$, in order to get to state $s_{x'}^{l_{end}}$ at time $h' + l_{end}$.

**Definition C.4** (time dependent reward class). Define $\mathcal{R}(n, A)$ to be the reward class. For $r \in \mathcal{R}(n, A)$,

$$
r_{x'}^{(h')}(s, a, h) = \begin{cases}
1, & s_1^0 & a_h \in A^{(s)} & \forall h < h' \\
0, & s_1^0 & a_h \notin A_h^{(\star)} & h = h' \\
1, & s_1^0 & a_h \in A_h^{(\star)} & h = h' \\
0, & s_1^0 & a_h \notin A^{(s)} & h < h' \\
0, & s_1^0 & \forall a_h & \forall h > h' \\
0, & s_x^{l_{end}} & \forall a_h & \forall h \neq h' + l_{end} & \forall x \\
0, & s_x^{l_{end}} & \forall a_h & h = h' + l_{end} & \forall x \neq x' \\
1, & s_x^{l_{end}} & \forall a_h & h = h' + l_{end} & x = x' \\
0, & s_x^l & \forall a_h & \forall h & \forall x & 0 < l < l_{end} \\
0, & s_x^{l_{end}+1} & \forall a_h & h \leq h' + l_{end} & \forall x \\
\nu^{(h')}[x], & s_x^{l_{end}+1} & \forall a_h & h > h' + l_{end},
\end{cases}
$$

where $\nu^{(h')}[x] \in [0, 1]$ is the reward at state $s_x^{l_{end}+1}$, with a specific selection of $h'$, and in general $\nu \in [0, 1]^{h' \times 2n}$. In addition, let $\tilde{h}$ be an upper bound for $h'$, and $x'$ to be the state number in layer $l_{end} = \log_2 n$ which has reward equal to 1 at time $h' + l_{end}$. Note that the optimal action at time $h = h'$ obeys $a_h \in A^{(u)} \cup A^{(d)}$.

In words, the new reward class incentivizes the learner to remain at the initial state until time $h'$, by providing rewards that are at least as large as those obtainable at the final layer (since $\nu^{(h')}[x] \in [0, 1]$). Moreover, the reward structure encourages the learner to reach a designated state, $s_{x'}^{l_{end}}$, at a specific time step by assigning a reward of 1 at time $h = h' + l_{end}$. It is important to note that the joint construction of the reward class in Definition C.4 and the transition class in Definition C.5 yields an MDP in which an optimal agent can collect all non-zero rewards up to layer $l_{end}$.

**Definition C.5** (time dependent transition class). Define $\mathcal{P}_{embd}$ to be the transition class. Now, for $\epsilon_0 = 1/8H$, and for $\mathbb{P} \in \mathcal{P}_{embd}(\epsilon_0)$:

$$
\begin{cases}
\mathbb{P}_h[s_1^0 & | s_1^0, a^{(s)}] & = 1 \\
\mathbb{P}_h[s_1^1 & | s_1^0, a^{(u)}] & = 1 \\
\mathbb{P}_h[s_2^1 & | s_1^0, a^{(d)}] & = 1 \\
\mathbb{P}_h[s_{2x-1}^{l+1} & | s_x^l, a^{(u)}] & = 1 & 0 < l < l_{end} \\
\mathbb{P}_h[s_{2x}^{l+1} & | s_x^l, a^{(d)}] & = 1 & 0 < l < l_{end} \\
\mathbb{P}_h[s_{x'}^{l_{end}+1} & | s_x^{l_{end}}, a] & = \mathbb{P}_h^{(x)}[x' | s_1^0, a] \\
\mathbb{P}_h[s_x^{l_{end}+1} & | s_x^{l_{end}+1}, \forall a] & = 1,
\end{cases}
$$

Where $a^{(s)}$, $a^{(u)}$ and $a^{(d)}$ are defined in Definition C.3, and $\mathbb{P}_h^{(x)}[s' = x' | s = s_1^0, a] \in \mathcal{P}_{single}(\epsilon_0)$ is the transition probability of a single state MDP at time $h$. The definition is an extension of "Description of Transition Class" in Jin et al. (2020b, Section D.3). In addition, we define that $\forall i \neq j$, and for all state $x$ within layer $l_{end}$, $\mathbb{P}_i^{(x)}(s' | s, a)$ and $\mathbb{P}_j^{(x)}(s' | s, a)$ are independent of each other.

In Definition C.5, we effectively construct an MDP that decomposes into $n\tilde{h}$ independent single-state learning problems. Specifically, each of the $n$ leaves of the deterministic binary tree defined by the MDP (corresponding to layers 1 through $l_{end}$) constitutes a single-state problem, while the time-dependent structure induces $\tilde{h}$ distinct instances of each of these $n$ leaves. In addition, we note that the optimal policy under Definitions C.2, C.4 and C.5 will collect the reward given in state $s_{x'}^{l_{end}}$ at time $h' + l_{end}$ with probability 1. This is the case, since the MDP is deterministic up to layer $l_{end}$, and the reward and transitions classes ensure that the maximum possible reward in this section of the MDP is obtainable, and is equal to $h' + 1$. Any other sequence of action other than the one that results in collecting the reward from state $s_{x'}^{l_{end}}$ at time $h' + l_{end}$, will yield lower cumulative reward.

**Definition C.6.** Here, we define two events that are relevant during the analysis given later in this section:

1. $\mathcal{A}_{(h)}$ is the event that $a_h = \arg\max_{a \in A} Q_h^\star(s_h, a)$.
2. $\mathcal{S}_{(h)}$ is the event that $s_h = s_1^0$.

In the following sections, the event $\mathcal{S}_{(h)} \cap \bar{\mathcal{A}}_{(h)}$ plays a key role in the analysis of the lower bound. In preparation for that, we now present the following lemma

**Lemma C.7.** *Under the events defined in Definition C.6, we have that $\forall j \neq i$ the events $\{\mathcal{S}_{(j)} \cap \bar{\mathcal{A}}_{(j)}\}$ and $\{\mathcal{S}_{(i)} \cap \bar{\mathcal{A}}_{(i)}\}$ are disjoint.*

*Proof.* Without loss of generality, let $j < i$. Now, we have that:

$$\mathbb{P}[\{\mathcal{S}_{(j)} \cap \bar{\mathcal{A}}_{(j)}\} \cap \{\mathcal{S}_{(i)} \cap \bar{\mathcal{A}}_{(i)}\}] \leq \mathbb{P}[\mathcal{S}_{(i)} \cap \bar{\mathcal{A}}_{(j)}] \leq \mathbb{P}[\mathcal{S}_{(i)} \mid \bar{\mathcal{A}}_{(j)}] = 0.$$

The last equality is true since given non-optimal action at time $j$ we have that $s_{j'} \neq s_1^0$ for all $j' > j$. $\qquad \square$

**Lemma C.8.** *Let $\pi$ be a policy that for $\epsilon \leq \frac{1}{5}$ and reward function $r_{x'}^{(h')}$ satisfies $\max_{\pi'} V_1^{\pi'}(s_1^0) - V_1^{\pi}(s_1^0) \leq \epsilon$ Then, the following statements hold*

$$\max_{\pi'} V_{h'}^{\pi'}(s_1^0) - V_{h'}^{\pi}(s_1^0) \leq \frac{\epsilon}{1-\epsilon} \leq \frac{1}{4} \quad and \quad \mathbb{P}[\mathcal{S}_{(h')} \cap \mathcal{A}_{(h')}] \geq 1 - \epsilon.$$

*Proof.* Using the events from Definition C.6, and for reward $r_{x'}^{(h')}$ we have:

$$
\begin{aligned}
\epsilon &\geq V_1^{\pi^\star}(s_1 = s_1^0) - V_1^{\pi}(s_1 = s_1^0) & \text{(by assumption)} \\
&= \sum_{h=1}^{H} \mathbb{E}_{\pi}\left[\left(V_h^{\pi^\star}(s_h) - Q_h^{\pi^\star}(s_h, a_h)\right)\right] & \text{(value difference lemma, see Puterman, 2014)} \\
&= \underbrace{\sum_{h=1}^{H} \mathbb{E}_{\pi}\left[\left(V_h^{\pi^\star}(s_h) - Q_h^{\pi^\star}(s_h, a_h)\right) \mid \mathcal{S}_{(h')} \cap \mathcal{A}_{(h')}\right] \cdot \mathbb{P}[\mathcal{S}_{(h')} \cap \mathcal{A}_{(h')}]}_{I} & \text{(total expectation)} \\
&\quad + \underbrace{\sum_{h=1}^{H} \mathbb{E}_{\pi}\left[\left(V_h^{\pi^\star}(s_h) - Q_h^{\pi^\star}(s_h, a_h)\right) \mid \overline{\mathcal{S}_{(h')} \cap \mathcal{A}_{(h')}}\right] \cdot \mathbb{P}[\overline{\mathcal{S}_{(h')} \cap \mathcal{A}_{(h')}}]}_{II}.
\end{aligned}
$$

**Expression ($I$) analysis**:

$$
\begin{aligned}
\epsilon &\geq \sum_{h=1}^{H} \mathbb{E}_{\pi}\left[\left(V_h^{\pi^\star}(s_h) - Q_h^{\pi^\star}(s_h, a_h)\right) \mid \mathcal{S}_{(h')} \cap \mathcal{A}_{(h')}\right] \cdot \mathbb{P}[\mathcal{S}_{(h')} \cap \mathcal{A}_{(h')}] \\
&= \underbrace{\left[V_{h'}^{\pi^\star}(s_1^0) - V_{h'}^{\pi}(s_1^0)\right]}_{:=\epsilon_{h'}} \cdot \mathbb{P}[\mathcal{S}_{(h')} \cap \mathcal{A}_{(h')}] \\
\Rightarrow \quad &\epsilon \geq \epsilon_{h'} \cdot \mathbb{P}[\mathcal{S}_{(h')} \cap \mathcal{A}_{(h')}].
\end{aligned}
$$

**Expressions ($II$) analysis**: First, we notice that since $\{\mathcal{S}_{(j)} \cap \bar{\mathcal{A}}_{(j)}\}$ and $\{\mathcal{S}_{(i)} \cap \bar{\mathcal{A}}_{(i)}\}$ are disjoint $\forall i \neq j$ (Lemma C.7), we have the following identity:

$$\overline{\mathcal{S}_{(h')} \cap \mathcal{A}_{(h')}} = \overline{\mathcal{S}}_{(h')} \cup \overline{\mathcal{A}}_{(h')} = \bigcup_{h \leq h'} \left\{\mathcal{S}_{(h)} \cap \bar{\mathcal{A}}_{(h)}\right\}.$$

Moreover, we adopt the following notation, where $\mathbb{I}_B$ denotes the indicator function of the event $B$:

$$\mathbb{E}[A \cdot \mathbb{I}_{\mathbb{B}}] := \mathbb{E}[A \mid B] \cdot \mathbb{P}[B]$$

Using the identity above, we have

$$
\epsilon \geq \sum_{h=1}^{H} \mathbb{E}_\pi \left[ \left( V_h^{\pi^\star}(s_h) - Q_h^{\pi^\star}(s_h, a_h) \right) \mid \overline{\mathcal{S}_{(h')} \cap \mathcal{A}_{(h')}} \right] \cdot \mathbb{P}[\overline{\mathcal{S}_{(h')} \cap \mathcal{A}_{(h')}}]
$$

$$
= \sum_{h=1}^{H} \mathbb{E}_\pi \left[ \left( V_h^{\pi^\star}(s_h) - Q_h^{\pi^\star}(s_h, a_h) \right) \cdot \mathbb{I}_{\overline{\mathcal{S}_{(h')} \cap \mathcal{A}_{(h')}}} \right] \qquad \text{(definition of } \mathbb{E}[A \cdot \mathbb{I}_B])
$$

$$
= \sum_{h=1}^{H} \mathbb{E}_\pi \left[ \left( V_h^{\pi^\star}(s_h) - Q_h^{\pi^\star}(s_h, a_h) \right) \cdot \sum_{j \leq h'} \mathbb{I}_{\mathcal{S}_{(j)} \cap \bar{\mathcal{A}}_{(j)}} \right] \qquad \text{(Lemma C.7)}
$$

$$
\geq \sum_{j \leq h'} \mathbb{E}_\pi \left[ \left( V_j^{\pi^\star}(s_j) - Q_j^{\pi^\star}(s_j, a_j) \right) \cdot \mathbb{I}_{\mathcal{S}_{(j)} \cap \bar{\mathcal{A}}_{(j)}} \right] \qquad \text{(linearity of expectation)}
$$

$$
= \sum_{j \leq h'} \mathbb{E}_\pi \left[ V_j^{\pi^\star}(s_j) - Q_j^{\pi^\star}(s_j, a_j) \mid \mathcal{S}_{(j)} \cap \bar{\mathcal{A}}_{(j)} \right] \cdot \mathbb{P}[\mathcal{S}_{(j)} \cap \bar{\mathcal{A}}_{(j)}] \qquad \text{(definition of } \mathbb{E}[A \cdot \mathbb{I}_B])
$$

$$
= \sum_{j \leq h'} \mathbb{E}_\pi \left[ Q_j^{\pi^\star}(s_1^0, \tilde{a}) - Q_j^{\pi^\star}(s_1^0, a_j) \right] \cdot \mathbb{P}[\mathcal{S}_{(j)} \cap \bar{\mathcal{A}}_{(j)}] \qquad (\tilde{a} = \arg\max_a Q_j^{\pi^\star}(s_1^0, a))
$$

$$
= \sum_{j \leq h'} \mathbb{E}_\pi \left[ \underbrace{r_{x'}^{(h')}(s_j = s_1^0, a_j = \tilde{a})}_{=1} - \underbrace{r_{x'}^{(h')}(s_j = s_1^0, a_j \neq \tilde{a})}_{=0} \right.
$$

$$
\left. + \underbrace{\mathbb{E}_{s' \sim \mathbb{P}_j(s' \mid s_j = s_1^0, a_j = \tilde{a})}[V_{j+1}^{\pi^\star}(s')] - \mathbb{E}_{s' \sim \mathbb{P}_j(s' \mid s_j = s_1^0, a_j \neq \tilde{a})}[V_{j+1}^{\pi^\star}(s')]}_{\geq 0 \ (\mathbb{I})} \right] \cdot \mathbb{P}[\mathcal{S}_{(j)} \cap \bar{\mathcal{A}}_{(j)}] \qquad (\star)
$$

$$
\geq \sum_{j \leq h'} \mathbb{P}[\mathcal{S}_{(j)} \cap \bar{\mathcal{A}}_{(j)}] \geq \mathbb{P}\left[ \bigcup_{j \leq h'} \{\mathcal{S}_{(j)} \cap \bar{\mathcal{A}}_{(j)}\} \right] := \mathbb{P}\left[ \overline{\mathcal{S}_{(h')} \cap \mathcal{A}_{(h')}} \right]. \qquad \text{(Union bound)}
$$

Here, $(\mathbb{I})$ in $(\star)$ follows directly from the construction of the reward class (Definition C.4) and the transition class (Definition C.5) - To reach state $s_{x'}^{l_{end}}$ at time $h' + l_{end}$ and obtain the associated reward, the learner must take optimal actions at every preceding step. Consequently, if a non-optimal action is taken at any time $j \leq h'$, the learner can no longer access this reward.

Note that conditioned on the event $\mathcal{S}_{(h')} \cap \mathcal{A}_{(h')}$, the optimal policy for the time dependent MDP (Definition C.5) is the same as for the non-time dependent MDP shown in (Jin et al., 2020b). In addition, we have the following relation between $\epsilon$ and $\epsilon_{h'}$:

$$
\mathbb{P}[\mathcal{S}_{(h')} \cap \mathcal{A}_{(h')}] \geq 1 - \epsilon \quad \Rightarrow \epsilon_{h'} \leq \frac{\epsilon}{1 - \epsilon} \leq \frac{1}{4}. \qquad \text{(for } \epsilon < \tfrac{1}{5})
$$

$\square$

Now, conditioning on the event that the policy leaves the initial state at time $h'$ (i.e., $\mathcal{S}_{(h')} \cap \mathcal{A}_{(h')}$), the MDP considered in this paper becomes identical to the one studied in Jin et al. (2020b), but with an effective horizon of $H - h'$. Consequently, we can directly invoke the guarantee provided by Jin et al. (2020b, Theorem 4.1) for a horizon of $H - h'$. Since $h' \leq \tilde{h} = \frac{H}{4}$, it follows that for every such $h'$, any $\epsilon_{h'}$-optimal algorithm must sample

$$
\Omega\left( \frac{|S|^2 |A| H^2}{\epsilon_{h'}^2} \right)
$$

trajectories.

**Lemma C.9.** *Fix some episode $t$. Let $h^\star$ be the time step in which the agent left state $s_1^0$ during the episode. In that case, the agent cannot collect samples from any transition kernel $\mathbb{P}_h$ where $h \neq h^\star + l_{end}$ during this episode.*

*Proof.* By the construction of the MDP and Definition C.5, we have that once the agent has left the initial state $s_1^0$, it takes a deterministic number of steps ($l_{end}$) to get to layer $l_{end}$ (which is the learning task layer). Furthermore, all of the states in layer $l_{end} + 1$, are absorbing. Importantly, this means that for every episode $t$, only one observation of $\mathbb{P}$ can be made (since the agent only visit layer $l_{end}$ once).

Thus, the construction of the MDP insures that for exit time $h^\star$ the agent will get to layer $l_{end}$ at time $h^\star + l_{end}$, and will not visit layer $l_{end}$ again in this episode. In addition, Definition C.5 states that for any $i \neq j$ the transition kernels $\mathbb{P}_i$ and $\mathbb{P}_j$ are independent of each other. Altogether, we have that the exit time from the initial state deterministically selects the specific transition dynamics that will be observed, which directly proves the lemma. $\square$

Lemma C.9 implies that we effectively have $\tilde{h} = \frac{H}{4}$ independent instances of the MDP from Jin et al. (2020b). Therefore, any $\epsilon$-optimal exploration algorithm is required to sample the MDP described in this paper for at least

$$\Omega\left(\frac{|S|^2|A|H^3}{\epsilon^2}\right)$$

trajectories.

## D. Additional lemmas

**Lemma D.1** (Chernoff bound). *Let* $X \sim Bin(n, p)$ *and let* $\mathcal{X} = \mathbb{E}[X]$. *For any* $0 < \zeta < 1$:
*Upper tail bound:*

$$\mathbb{P}(X \geq (1 + \zeta)\mathcal{X}) \leq exp\left\{-\frac{\zeta^2\mathcal{X}}{3}\right\}$$

*Lower tail bound:*

$$\mathbb{P}(X \geq (1 - \zeta)\mathcal{X}) \leq exp\left\{-\frac{\zeta^2\mathcal{X}}{2}\right\}$$

**Lemma D.2** (Self normalized sum). *Let* $\{a_i\}_{i=1}^n$ *be a sequence of numbers:*
*Define* $S_i = \sum_{j=1}^i a_j$, *for* $a_i \in \mathbb{R}$:

$$\sum_{i=1}^n \frac{a_i}{\sum_{j=1}^i a_j} \leq 1 + \log\left(\frac{S_n}{S_1}\right)$$

*Define* $S_i = 1 + \sum_{j=1}^{i-1} a_j$, *for* $a_i \in (0, 1]$:

$$\sum_{i=1}^n \frac{a_i}{1 + \sum_{j=1}^{i-1} a_j} \leq 2\log\left(1 + \sum_{i=1}^n a_i\right)$$

*Proof.* Separately for each part.
For $a_i \in (0, 1]$:
$\forall x \in [1, 2]$ we have that $x - 1 \leq 2\log x$, and note that:

$$\frac{S_{i+1}}{S_i} = \frac{a_i + S_i}{S_i} = \frac{a_i}{S_i} + 1 \leq 2$$

$$\Rightarrow \sum_{i=1}^n \frac{a_i}{1 + \sum_{j=1}^{i-1} a_j} = \sum_{i=1}^n \frac{S_{i+1} - S_i}{S_i} = \sum_{i=1}^n \left(\frac{S_{i+1}}{S_i} - 1\right) \leq 2\sum_{i=1}^n \log\frac{S_{i+1}}{S_i} = 2\log\frac{S_{n+1}}{S_1} = 2\log\left(1 + \sum_{i=1}^n a_i\right)$$

For $a_i \in \mathbb{R}$:
$\forall x \in \mathbb{R}$ we have that $1 - x \leq \log(1/x)$:

$$\sum_{i=1}^n \frac{a_i}{\sum_{j=1}^i a_j} = 1 + \sum_{i=2}^n \frac{S_i - S_{i-1}}{S_i} = 1 + \sum_{i=2}^n \left(1 - \frac{S_{i-1}}{S_i}\right) \leq 1 + \sum_{i=2}^n \log\frac{S_i}{S_{i-1}} = 1 + \log\frac{S_n}{S_1}.$$

$\square$

**Lemma D.3** (Freedman's inequality - Lemma C.4 in Amortila et al.). *Let $\{X_t\}_{t=1}^{T}$ be a sequence of random variables adapted to a filtration $\{\mathcal{F}_t\}_{t=1}^{T}$, and define $\mathbb{E}_t[\cdot] := \mathbb{E}[\cdot \mid \mathcal{F}_t]$. If $0 \leq X_t \leq R$ almost surely, then with probability at least $1 - \delta$,*

$$\sum_{t=1}^{T} X_t \leq \frac{3}{2} \sum_{t=1}^{T} \mathbb{E}_{t-1}[X_t] + 4R \ln(2\delta^{-1})$$

*and*

$$\sum_{t=1}^{T} \mathbb{E}_{t-1}[X_t] \leq 2 \sum_{t=1}^{T} X_t + 8R \ln(2\delta^{-1})$$

**Lemma D.4** (Weissman bound). *Let $P(\cdot)$ be a distribution over $m$ elements, and let $\hat{P}_t(\cdot)$ be the empirical distribution defined by $t$ iid samples from $P(\cdot)$. Then, with probability at least $1 - \delta$*

$$||\hat{P}_t(\cdot) - P(\cdot)||_1 \leq 2\sqrt{\frac{m \log(\delta^{-1})}{t}}$$

*This is a direct result from **Theorem 2.1** in (Weissman et al., 2003).*

**Lemma D.5.** *Let $V^{\mathbb{P},\pi}(s_1)$ be the value function under dynamics $\mathbb{P}$ and policy $\pi$, and $|r|_\infty \leq B$. Define $N(s_h, a_h)$ to be the number of samples given for $(s_h, a_h)$, and $H$ to be the horizon of the settings. In that case, w.p. at least $1 - \delta$, $\forall \pi$ the following holds:*

$$\left| V_1^{\mathbb{P},\pi}(s_1) - V_1^{\hat{\mathbb{P}},\pi}(s_1) \right| \leq 3 \cdot \sqrt{HB \cdot V^{\mathbb{P},\pi}(s_1)} \cdot \sqrt{\mathbb{E} \sum_{h=1}^{H} \frac{C_1|S|}{N(s_h, a_h)} + \mathbb{E} \sum_{h=1}^{H} \frac{6C_1 H^2 |S| B}{N(s_h, a_h)}}$$

*Where after using the union bound for $|S|, |A|, H$ and $T$, and using **Corollary 5** in (Maurer & Pontil), we have $C_1 = 5\ell_0$. Note the change of $N(s_h, a_h) - 1$ to $N(s_h, a_h)$ compared to (Maurer & Pontil), and the doubling of the constant since $\frac{1}{N(s_h, a_h) - 1} \leq \frac{2}{N(s_h, a_h)}$ for $N(s_h, a_h) \geq 2$.*

*Proof.* We start by decomposing the value functions difference to 3 components, as seen below

$$\left| V_h^\pi(s) - \hat{V}_h^\pi(s) \right|$$

$$= \left| \left( r(s, \pi(s)) + P_{h,s,\pi(s)} V_{h+1}^\pi(s) \right) - \left( r(s, \pi(s)) + \hat{P}_{h,s,\pi(s)} \hat{V}_{h+1}^\pi(s) \right) \right|$$

$$= \left| P_{h,s,\pi(s)} V_{h+1}^\pi(s) - \hat{P}_{h,s,\pi(s)} \hat{V}_{h+1}^\pi(s) \right|$$

$$\leq \underbrace{\left| (P_{h,s,\pi(s)} - \hat{P}_{h,s,\pi(s)}) V_{h+1}^\pi(s) \right|}_{(I)} + \underbrace{\left| (P_{h,s,\pi(s)} - \hat{P}_{h,s,\pi(s)})(V_{h+1}^\pi - \hat{V}_{h+1}^\pi)(s) \right|}_{(II)} + \underbrace{\left| P_{h,s,\pi(s)}(V_{h+1}^\pi - \hat{V}_{h+1}^\pi)(s) \right|}_{(III)}$$

**Analysis of component $I$**

$$\left| (P_{h,s,\pi(s)} - \hat{P}_{h,s,\pi(s)}) V_{h+1}^\pi(s) \right| \leq \sqrt{\frac{C_1|S| \operatorname{Var}_{P_{h,s,\pi(s)}}(V_{h+1}^\pi)}{N(s, \pi(s))}} + \frac{C_1|S|HB}{N(s, \pi(s))} \qquad \text{(Maurer \& Pontil, Theorem 4)}$$

**Analysis of component $II$**

$$
\left| (P_{h,s,\pi(s)} - \hat{P}_{h,s,\pi(s)})(V_{h+1}^\pi - \hat{V}_{h+1}^\pi)(s) \right|
$$

$$
\leq \sum_{s'} \sqrt{\frac{C_1 P_h(s'|s,\pi(s))}{N(s,\pi(s))}} \cdot \left| V_{h+1}^\pi(s') - \hat{V}_{h+1}^\pi(s) \right| + \frac{C_1 |S| H B}{N(s,\pi(s))} \qquad \text{(Maurer \& Pontil, Theorem 4)}
$$

$$
\leq \sum_{s'} \left\{ \frac{4 C_1 H^2 B}{N(s,\pi(s))} + \frac{1}{H^2 B} \cdot P_h(s'|s,\pi(s)) \cdot \left( V_{h+1}^\pi(s') - \hat{V}_{h+1}^\pi(s) \right)^2 \right\} + \frac{C_1 |S| H B}{N(s,\pi(s))} \qquad \text{(AM-GM)}
$$

$$
\leq \frac{4 C_1 H^2 |S| B}{N(s,\pi(s))} + \frac{1}{H} \cdot \sum_{s'} \left\{ P_h(s'|s,\pi(s)) \cdot \left( V_{h+1}^\pi(s') - \hat{V}_{h+1}^\pi(s) \right) \right\} + \frac{C_1 |S| H B}{N(s,\pi(s))}
$$

**Analysis of component $III$**

$$
\left| P_{h,s,\pi(s)}(V_{h+1}^\pi - \hat{V}_{h+1}^\pi)(s) \right| \leq \sum_{s'} P_h(s,\pi(s)) \left| V_{h+1}^\pi(s') - \hat{V}_{h+1}^\pi(s') \right|
$$

Combining the results obtained from the analysis above, we have that

$$
\left| V_h^\pi(s) - \hat{V}_h^\pi(s) \right| \leq \sqrt{\frac{C_1 |S| \operatorname{Var}_{P_{h,s,\pi(s)}}(V_{h+1}^\pi)}{N(s,\pi(s))}} + \frac{2 C_1 |S| H B + 4 C_1 H^2 |S| B}{N(s,\pi(s))}
$$

$$
+ \left( 1 + \frac{1}{H} \right) \cdot P_{h,s,\pi(s)} |V_{h+1}^\pi(s_1) - \hat{V}_{h+1}^\pi(s_1)|
$$

And unrolling the recursive equation to $h = 1$ yields

$$
\left| V_1^\pi(s_1) - \hat{V}_1^\pi(s_1) \right| \leq \mathbb{E}^{\pi,P} \left\{ \sum_{h=1}^H \underbrace{\left( 1 + \frac{1}{H} \right)^{H-h}}_{\leq 3} \left( \sqrt{\frac{C_1 |S| \operatorname{Var}_{P_{h,s,\pi(s_h)}}(V_{h+1}^\pi)}{N(s_h,a_h)}} + \frac{2 C_1 |S| H B + 4 C_1 H^2 |S| B}{N(s_h,a_h)} \right) \right\}
$$

$$
\leq 3 \sqrt{\mathbb{E} \sum_{h=1}^H \operatorname{Var}_{P_{h,s,\pi(s_h)}}(V_{h+1}^\pi)} \cdot \sqrt{\mathbb{E} \sum_{h=1}^H \frac{C_1 |S|}{N(s_h,a_h)} + \mathbb{E} \sum_{h=1}^H \frac{6 C_1 H^2 |S| B}{N(s_h,a_h)}}
$$

$$
\leq 3 \sqrt{B H V_1(s_1)} \cdot \sqrt{\mathbb{E} \sum_{h=1}^H \frac{C_1 |S|}{N(s_h,a_h)} + \mathbb{E} \sum_{h=1}^H \frac{6 C_1 H^2 |S| B}{N(s_h,a_h)}},
$$

where the final inequality is by Lemma B.7. $\qquad \square$

