# OpenReview forum: "Improved Bounds for Reward-Agnostic and Reward-Free Exploration"
_ICML.cc/2026/Conference — ICML 2026 regular_

### Official Review · Reviewer_smz6 · 2026-03-09

**Soundness:** 3
**Presentation:** 3
**Significance:** 2
**Originality:** 2
**Overall Recommendation:** 4
**Confidence:** 2

**Summary:**

This paper studies reward-free and -agnostic exploration (RAE) with time-inhomogeneous reward functions for tabular MDPs. In particular, this paper presents the following results: (1) the improved sample complexity bound for RAE in the lower order term, and (2) the sample complexity lower bound for time-inhomogeneous reward-free exploration (RFE).

For (1), this paper proposes an algorithm for constructing an exploratory policy, which is an online MDP algorithm with a sophisticatedly-designed reward function sequence. For (2), this paper extends the lower bound argument for time-homogeneous RFE to time-inhomogeneous RFE.

**Compliance With Llm Reviewing Policy:**

Affirmed.

**Final Justification:**

The technical flaw in the analysis has been addressed in the author response. Therefore, I raise my scores for soundness and overall recommendation.

**Key Questions For Authors:**

To derive Eq. (18), the paper claims that $\hat V_{h+1}$ is independent of $\hat P_h$, so Lemma B.7 is applicable. My question is about whether they are indeed independent. In particular, $\hat V_{h+1}$ is directly computed by $\hat P_{h+1}$, which is determined by $N_{h+1}(s,a)$. Since it is the number of visits $(s,a)$ in step $h+1$, it may be associated with the event that the trajectory visits $(s',a')$ in step $h$ for some $s',a'$. In summary, I'm not sure that $\hat P_{h+1}$ and $\hat P_h$ are independent. In that sense, I'm concerned with $\hat V_{h+1}$ may depend on $\hat P_h$, even when $\hat P_h$ is not directly used for computing $\hat V_{h+1}$.

**Limitations:**

yes

**Strengths And Weaknesses:**

**Strength**

1. For RAE, the paper proposes an improved sample complexity bound of $\tilde O(H^3 AS/\epsilon^2 + H^4 AS^2/\epsilon)$, which improves upon the previous work of $\tilde O(H^3 AS/\epsilon^2 + H^6 A^4S^4/\epsilon)$.
2. The algorithmic novelty comes from Algorithm 2, which constructs an exploratory policy. In particular, it is based on an online MDP algorithm whose reward function is defined as $\frac{1}{c + N_t(s,a,h)}$, which is designed to promote efficient exploration.
3. For RFE, the paper presents a tighter lower bound of $\Omega(H^3 AS^2/\epsilon^2)$. The previous lower bound in this regime, which is $\Omega(H^2 AS^2/\epsilon^2)$, is limited to the time-homogeneous setting. The paper extends the lower bound argument in [Jin et al. 2020]---lower bound for the time-homogeneous setting---to the time-inhomogeneous setting.

**Weakness**

1. The improvement of regret bound for RAE lies in the lower order term. Since the higher order term still remains the same as that in [Li et al. 2024], this improvement could be viewed as incremental.
2. Although the proposed algorithm improves the sample complexity bound for RAE, its sample complexity bound for RFE becomes worse than that of [Menard et al. 2021], due to the additional term of $\frac{H^4 AS^2}{\epsilon}$.
3. There are some potential logical flaws in the analysis. Please, see the questions for details.

**Refs**

[Jin et al. 2020] Reward-Free Exploration for Reinforcement Learning, ICML 2020.

[Menard et al. 2021] Fast active learning for pure exploration in reinforcement learning, ICML 2021.

[Li et al. 2024] Minimax-Optimal Reward-Agnostic Exploration in Reinforcement Learning, COLT 2024.

---

> ### Author Rebuttal · Authors · 2026-03-29
>
> We sincerely thank the reviewer for their time, constructive feedback, and thoughtful evaluation of our work. We appreciate the opportunity to clarify our theoretical contributions and address the specific concerns raised. Below, we provide detailed responses to each of your points.
>
> **Degradation in performance for RFE**
>
> We apologize if this was not emphasized enough. Our primary contribution to the RFE setting is not an improvement on the upper bound, but rather establishing the first tight lower bound of $\Omega(\frac{|S|^2|A|H^3}{\epsilon^2})$ for time-inhomogeneous MDPs. This strictly proves that prior algorithms (like [Ménard et al., 2021] (" Fast active learning for pure exploration in reinforcement learning")) are indeed minimax optimal and resolves a standing conjecture in the field. Our goal was to obtain an optimal bound for RAE, though unlike [Ménard et al., 2021] (" Fast active learning for pure exploration in reinforcement learning"), our algorithm works for both RFE and RAE. The result for the RFE setting was simply provided for completeness. We will discuss this coherently in the camera-ready version.
>
>
> **The derivation of eq. 18**
>
> We sincerely thank the reviewer for their careful reading and for identifying this subtle but important technicality. You are absolutely correct: because our data is collected via sequential full trajectories (algorithm 3), the empirical transitions at step $h$ inherently influence the state visitation counts at step $h+1$. Consequently, the claim of strict statistical independence between  $\hat{P}\_{h}$  and  $\hat{V}\_{h+1}$  as currently written in the manuscript is technically incomplete.
>
>
> To rigorously resolve this dependency, and to preserve our optimal high-order sample complexity bounds, we will integrate the *two-fold subsampling* technique introduced by [Li et al. (2024a)] ("Settling the sample complexity of model-based offline reinforcement learning"). This method elegantly decouples the sequential dependency of the trajectories by separating the data used to determine visitation quotas from the data used to estimate the transitions.
>
>
> The algorithm outlines as follow (taken directly from [Li et al. (2024a)] ("Settling the sample complexity of model-based offline reinforcement learning")):
>
> * First, randomly split the dataset into two halves $\mathcal{D}^{main}$ and $\mathcal{D}^{aux}$, where $\mathcal{D}^{main}$ consists of $N\_{h}^{main}(s,a)$ sample transitions from state action $s, a$ at step h.
>
> * For each $(s, a,h) \in S \times A \times [H]$, we use the dataset $\mathcal{D}^{aux}$ to construct a high-probability lower bound $N\_{h}^{trim}(s,a)$ on $N\_{h}^{main}(s,a)$, and then subsample $N\_{h}^{trim}(s)$ sample transitions w.r.t. $(s, a,h)$ from $\mathcal{D}^{main}$ ; this results in a new subsampled dataset $\mathcal{D}^{trim}$.
>
>
> In the camera-ready version of the paper, we will explicitly mention the use of this procedure, and update the corresponding proofs in Appendix B. In detail, we will implement the following steps:
>
>
> * **Algorithmic Update:** During Algorithm 3, after data collection, we will simply call the “two-fold subsampling” procedure (Algorithm 3, [Li et al. (2024a)] ("Settling the sample complexity of model-based offline reinforcement learning")) to alter our dataset, obtaining $\mathcal{D}^{trim}$ as explained above. Thereafter, dynamics estimation will be done based on $\mathcal{D}^{trim}$. Note that the call to Li et al.’s procedure is done in a completely black box fashion.
>
> * **Theoretical Update (Ensuring Independence):** In the final version, we will add a short explanation justifying the “two-fold subsampling” procedure correctness, and specifically reference the detailed explanation given at [Li et al. (2024a)] ("Settling the sample complexity of model-based offline reinforcement learning"). The use of this procedure cleanly satisfies the requirements for Lemma B.7 and eq. 18 therein.
>
> * **Theoretical Update (Preserving Complexity):** Finally, using Lemma 3 from [Li et al. (2024a)] ("Settling the sample complexity of model-based offline reinforcement learning"), we have that $N\_{h}(s, a) \ge N\_h^{trim}(s, a) \ge \frac{N\mu^{T}\_{h}(s, a)}{16}$, for $\mu^{T}\_{h}(s, a) \ge O(1 / N)$, up to logarithmic factors. This is  true for all significant $s, a, h$ since $\mu^{T}\_{h}(s, a) \ge \omega = O\left(\frac{\epsilon}{poly(|S|, |A|, H)}\right) \ge O\left(\frac{\epsilon^2}{poly(|S|, |A|, H)}\right) = O(1 / N)$, for small enough $\epsilon$. This allows as to modify the last condition in the high probability events $E\_{RAE}$ and $E\_{RFE}$ to be $N\_h^{trim}(s, a) \ge \frac{N\mu^{T}\_{h}(s, a)}{16}$, thus only effecting the final bound by a constant multiplicative factor.
>
>
> We believe this standard theoretical patch fully addresses the dependency concern, and we deeply appreciate the reviewer's sharp eye in helping us fortify the mathematical rigor of the paper.

---

> > ### Author Rebuttal · Reviewer_smz6 · 2026-04-02
> >
> > Thanks for the detailed author's response.
> >
> > The main concern in the analysis (also the main reason for the low score) is addressed. Specifically, it is reasonable that adopting the technique from Li et al. resolves the dependency issue. This change does not affect the paper's main contributions. I expect that the algorithm and analysis would be updated in the camera-ready version. Accordingly, the score will be updated.

---

> > > ### Author Response · Authors · 2026-04-04
> > >
> > > Thank you for your response, and for updating the score.

---

### Official Review · Reviewer_dv32 · 2026-03-10

**Soundness:** 4
**Presentation:** 3
**Significance:** 3
**Originality:** 3
**Overall Recommendation:** 4
**Confidence:** 4

**Summary:**

The paper studies exploration in episodic finite-horizon tabular time-inhomogeneous MDPs when no external rewards are observed during exploration. For RFE, the work provides a tight lower bound for time-inhomogeneous RFE and presents an algorithm that achieves the minimax upper bound (up to logarithmic factors). For RAE, the claimed sample complexity significantly improves the low-order dependence to $\frac{H^4|A||S|^2}{\epsilon}$.

**Compliance With Llm Reviewing Policy:**

Affirmed.

**Ethical Review Concerns:**

Thank you for the detailed and clarifying response. The explanations regarding the role of the designed reward sequence and the use of a single online MDP procedure are helpful and address my main concerns. The clarification on computational aspects and the positioning relative to prior work (especially for RFE) is also appreciated. Overall, I will maintain the positive score.

**Final Justification:**

The author has addressed my main concerns. I think this work is meaningful and improved the understanding on the reward-free problem framework. I maintain my positive score.

**Key Questions For Authors:**

1. What is the runtime per episode for Algorithm 5's OMD step under constraints (13) and (14)? Is it “just polynomial”? Can it be solved efficiently in practice?

2. In the RAE, what constrains the dependence of sample complexity on $H$ from reaching optimal $H^2$?

3. In table 1, reward-free setting, what is the concrete improvement between this paper and Menard et al. (2021)'s work, since the dominant upper bound term remains the same, with an additional low-order term $\frac{H^4|A||S|^2}{\epsilon}$.

**Limitations:**

yes

**Strengths And Weaknesses:**

**Strengths**
1. This work addresses an important theoretical gap, as reward-free/agnostic exploration is a foundational question in RL theory, and the paper directly targets minimax sample complexity in finite-horizon tabular settings.

2. The low-order sample complexity dependence on $H, |A|, |S|$ is significantly reduced by reusing samples for both RFE and RAE, supported by technically solid analysis.

3. The sample complexity $\Omega(\frac{|S|^2|A|H^3}{\epsilon^2})$ is the first tight lower bound established for time-inhomogeneous RFE.



**Weaknesses**
1. The paper contains no experiments. Providing an additional baseline comparison with the algorithm in Li et al. (2024b) would greatly strengthen the credibility.

2. It's difficult for me to clearly see how the carefully designed reward sequence facilitated better analytical results compared to previous work. Including a proof sketch that contrasts the core techniques with prior work would greatly enhance readability.

---

> ### Author Rebuttal · Authors · 2026-03-29
>
> We sincerely thank the reviewer for their time, constructive feedback, and thoughtful evaluation of our work. We appreciate the opportunity to clarify our theoretical contributions and address the specific concerns raised. Below, we provide detailed responses to each of your points.
>
> * **Empirical Results:** We would like to respectfully clarify that the primary contribution of this paper is entirely theoretical. As is standard for theoretical submissions to ICML, our focus here is solely on establishing rigorous mathematical guarantees and foundational insights. While we agree that empirical validation is an exciting next step, we believe the theoretical results presented here are substantial enough to stand on their own. We leave the simulative and empirical validation of these bounds for future work.
>
> * **Carefully designed reward sequence:** The main novelty over previous works is use of an online algorithm in order to explore the MDP, rather than invoking an expensive stochastic no-regret algorithm at each iteration. This method allowed us to reduce the low-order term in the bound significantly, by re-using data collected between episodes. The reward sequence is simply designed to create efficient exploration using our method (see the end of Section 3: “This section motivates… … property up to logarithmic factors”).
>
> * **OMD Runtime / Practical Efficiency:** The OMD step in Algorithm 5 represents a convex optimization problem with linear constraints. As such, and as discussed at Section A.1. at [Jin et al., 2020a] ("Learning adversarial markov decision processes with bandit feedback and unknown transition"), It can be solved efficiently in polynomial time (using online convex optimization methods). Furthermore, to make it highly efficient in practice, it can be reformulated into a dual problem with only non-negativity constraints.
>
> * **Optimal H dependency for RAE:** The lower bound for RAE provided by [Zhang et al., 2020] (" Task-agnostic exploration in reinforcement learning"), is for the time homogeneous setting. The relevant lower bound to the non-homogeneous setting is the one given by [Domingues et al. (2021)] ("Episodic reinforcement learning in finite mdps: Minimax lower bounds revisited"). Similarly to the H dependency in the RFE, there is a factor H difference between the time dependent, and time independent settings. This means that the upper bound proved in our paper is actually minimax optimal with regard to the leading term.
>
> * **RFE Improvement over Ménard et al.:** We apologize if this was not emphasized enough. Our primary contribution to the RFE setting is not an improvement on the upper bound, but rather establishing the first tight lower bound of $\Omega(\frac{|S|^2|A|H^3}{\epsilon^2})$ for time-inhomogeneous MDPs. This strictly proves that prior algorithms (like [Ménard et al., 2021] (" Fast active learning for pure exploration in reinforcement learning")) are indeed minimax optimal and resolves a standing conjecture in the field. The low-order term in our upper bound is a necessary structural tradeoff for providing a single, unified meta-algorithm, and the result for the RFE setting was provided for clarity and completeness. We will explain this coherently in the camera-ready version.

---

> > ### Author Rebuttal · Reviewer_dv32 · 2026-04-02
> >
> > Thank you for the detailed and clarifying response. The explanations regarding the role of the designed reward sequence and the use of a single online MDP procedure are helpful and address my main concerns. The clarification on computational aspects and the positioning relative to prior work (especially for RFE) is also appreciated. Overall, I will maintain the positive score.

---

> > > ### Author Response · Authors · 2026-04-04
> > >
> > > Thank you for your response.

---

### Official Review · Reviewer_siuq · 2026-03-12

**Soundness:** 4
**Presentation:** 4
**Significance:** 3
**Originality:** 3
**Overall Recommendation:** 5
**Confidence:** 2

**Summary:**

The paper considers reward-agnostic and reward-free exploration in episodic finite-horizon MDPs. It proposes new algorithms together with a new analysis framework based on occupancy measures and optimistic planning, and derives improved sample-complexity bounds with better dependence on the horizon $H$ than prior work.

**Compliance With Llm Reviewing Policy:**

Affirmed.

**Final Justification:**

After reading the authors' rebuttal and the discussion with other reviewers I choose to keep my score.

**Key Questions For Authors:**

I would appreciate it if the authors could comment on whether, and to what extent, the results in the paper can be extended to non-episodic MDP settings, such as standard average-reward and discounted-reward MDPs under suitable connectivity assumptions.

**Limitations:**

The limitations are clearly discussed in the paper. I also do not see potential negative societal impact specific to this work.

**Strengths And Weaknesses:**

The paper is clearly written and logically coherent. The overall structure is easy to follow, the technical development is well organized, and the main message is communicated clearly. A particular strength of the paper is that it does not merely sharpen existing guarantees, but does so through new algorithms and a new line of analysis. This makes the contribution more substantial than a routine refinement of prior bounds. The occupancy-measure-based perspective appears interesting, and the resulting improvement in the dependence on the horizon $H$ seems meaningful.

Overall, my impression is positive: the paper is well presented, technically solid, and appears to make a meaningful theoretical contribution. While I am not an expert specifically in the theory of exploration, the results and proofs appear correct to me, and the proposed analysis route seems novel and interesting.

---

> ### Author Rebuttal · Authors · 2026-03-29
>
> We sincerely thank the reviewer for their time, constructive feedback, and thoughtful evaluation of our work. Thank you for recognizing the merits of our paper. Below, we provide a detailed response to your question.
>
> **Extension to Non-Episodic Settings:** This is an excellent theoretical question for future research. Our current framework is heavily tailored to finite-horizon episodic MDPs, and the extension for generic non-episodic settings will require a complete redesign of the algorithm. Specifically, the pessimistic offline planning phase (Algorithm 4) fundamentally relies on backward induction starting from $h = H$. Furthermore, our lower bound construction depends strictly on the finite horizon, utilizing time-dependent transition dynamics and specific exit times up to $O(H)$.

---

> > ### Author Rebuttal · Reviewer_siuq · 2026-04-03
> >
> > I thank the authors for answering my question.

---

> > > ### Author Response · Authors · 2026-04-04
> > >
> > > Thank you for your response.

---

### Official Review · Reviewer_BR5a · 2026-03-12

**Soundness:** 3
**Presentation:** 2
**Significance:** 2
**Originality:** 2
**Overall Recommendation:** 4
**Confidence:** 3

**Summary:**

This paper studies reward-free exploration in episodic finite horizon tabular MDPs. Overall, the main contribution concerns two results: first, a three-stage algorithm that constructs an exploration policy via a single online-MDP run on carefully designed intrinsic rewards, then estimates the dynamics in a pessimistic way (achieving  a bound of order $O(H^3 A S^2/\epsilon^2 + H^4A S^2/\epsilon$ in the reward-free setting). second, a tighter  lower bound for reward-free exploration $O(S^2 A H^3/\epsilon^2)$.  The algorithmic novelty is to replace repeated no-regret subroutines with a single online-MDP procedure driven by intrinsic rewards. The offline planning stage then uses pessimistic model-based RL on data collected by the exploration policy.  No empirical results are presented.

**Compliance With Llm Reviewing Policy:**

Affirmed.

**Final Justification:**

The paper makes two contributions. The tight lower bound  for time-inhomogeneous RFE is the stronger result, and I consider this a solid contribution to the RL theory literature.
The upper-bound improvement is limited to the low-order term, as I noted in my original review. However, the algorithmic technique seems novel and may inspire future work.
Regarding the proof issue identified by smz6, it may only affects constant factors, since the results target order-optimal bounds. This possibly does not affect the main contributions.
My residual concerns are: (1) empirically validate the algorithm; (2) the related work section to be expanded; (3) the computational complexity should be stated explicitly in the main text. The authors have committed to addressing (2)-(3) in the camera-ready.

Overall, the lower bound result plus the novel algorithmic technique justify my weak accept score.

**Key Questions For Authors:**

What is the computational complexity of the algorithm?

**Limitations:**

Limitations should be better discussed: restriction to tabular MDPs; the absence of empirical results.

**Strengths And Weaknesses:**

- Soundness: the paper is technically sound, but there are some sloppy parts in the writeup in terms of typo (see below).
- Presentation: The paper is generally well structured. The high-level idea of decomposing the problem into exploration-policy creation and pessimistic offline planning is easy to follow, and the convex-optimization/log-barrier interpretation is helpful. There are still some typos, see theorem 5.2 about $V_1^\pi$ and lemma 4.5 about $\mu_t$.
- Significance: The lower-bound result is significant. If correct (i did not check the proof),  it is a good contribution.  The upper-bound side is somewhat more modest in significance, because the leading-order terms were known. However, there are some limitations. Results are specialized to tabular MDPs, and and there are no empirical results validating the claims.

- Originality: I believe the most original component is the exploration-policy construction, using one online-MDP procedure over adaptive intrisic rewards to approximate the desired occupancy-measure coverage objective.
- Limitations should be better discussed: restriction to tabular MDPs; the absence of empirical results.
- There is related work that is not discussed/considered by the authors. The authors should discuss differences and compare the results. The paper should mention On the Statistical Efficiency of Reward-Free Exploration in Non-Linear RL (NeurIPS 2022), Active Coverage for PAC Reinforcement Learning (COLT 2023), which is directly about coverage-oriented exploration objectives. More closely related there are some recent papers on pure exploration for best policy identification in MDPs in the multiple reward setting. See Multi-Reward Best Policy Identification (NeurIPS 2024), and Adaptive Exploration for Multi-Reward Multi-Policy Evaluation  (ICML 2025).
- The authors should also fix the references. some manuscripts are already published as proceedings, and should not be listed as preprints.

---

> ### Author Rebuttal · Authors · 2026-03-29
>
> We sincerely thank the reviewer for their time, constructive feedback, and thoughtful evaluation of our work. We appreciate the opportunity to clarify our theoretical contributions and address the specific concerns raised. Below, we provide detailed responses to each of your points.
>
> * **Empirical Results:** We would like to respectfully clarify that the primary contribution of this paper is entirely theoretical. As is standard for theoretical submissions to ICML, our focus here is solely on establishing rigorous mathematical guarantees and foundational insights. While we agree that empirical validation is an exciting next step, we believe the theoretical results presented here are substantial enough to stand on their own. We will discuss this point in the final version of our work.
>
> * **Typos:** We appreciate your thorough reading. We will correct the typos in Theorem 5.2 and Lemma 4.5.
>
> * **Restriction to tabular MDPs:** The focus of this work is indeed tabular MDPs. Our guiding philosophy is to first understand tabular MDPs as comprehensively as possible, before moving forward to more complex function approximation.  In the final version, we will mention non-tabular MDPs as an important avenue for future work.
>
> * **Missing  related work:** Thank you for the pointers. We will add and discuss the missing references in the final version of our work.
>
> * **Citations:** Thank you for catching that! we will update the preprint references to their published proceedings.
>
> * **Computational Complexity:** The OMD step in Algorithm 5 represents a convex optimization problem with linear constraints. As such, and as discussed at Section A.1. at [Jin et al., 2020a] ("Learning adversarial markov decision processes with bandit feedback and unknown transition"), It can be solved efficiently in polynomial time (using online convex optimization methods). Furthermore, to make it highly efficient in practice, it can be reformulated into a dual problem with only non-negativity constraints.

---

> > ### Author Rebuttal · Reviewer_BR5a · 2026-04-03
> >
> > Thank you for the rebuttal.
> >
> > -  I still think the paper should explicitly state its computational complexity rather than only saying the OMD subproblem is polynomial-time, since the optimization is over augmented occupancy measures and may be large in practice.
> > - I also continue to think the related-work section needs to be expanded, especially around active coverage and recent multi-reward   papers.
> > -  Scientifically, my view is mostly unchanged: the upper-bound contribution is mainly a lower-order-term improvement, while the tighter reward-free lower bound is the stronger part of the paper.
> > - I appreciate the commitment to fix the clear typos and the bibliography.
> >  - I agree that the paper is theoretical, but experiments would substantially improve the paper. For this reason, I keep my score.

---

> > > ### Author Response · Authors · 2026-04-04
> > >
> > > Thank you for your feedback. Our response is as follows:
> > >
> > > *“paper should explicitly state its computational complexity”*
> > > \
> > > We appreciate your opinion on this. The computational complexity of one OMD step is $O(|S|^6 |A|^3 H^3)$.  This is the case since solving the OMD step is a convex optimization program of dimension $n$, with $m$ linear inequality constraints. As such, we can solve it with newton’s method in $O((m+n)^3)$ time in the worst case. In our case $m=O(|S|^2 |A| H)$ and $n=O(|S|^2 |A| H)$, so overall we are talking about $O(|S|^6 |A|^3 H^3)$ time. We will add this in the final version.
> > >
> > > *“related-work section needs to be expanded”*
> > > \
> > > We will update the related work section to include the active coverage literature.
> > >
> > > *“the upper-bound contribution is mainly a lower-order-term improvement”*
> > > \
> > > While the reviewer is correct that the improvement in the bound is in the low-order term, we emphasize that it is attained using an entirely novel algorithmic technique. We believe this is an important addition to the literature on exploration in RL, and that future work will build on this to attain minimax-optimal bounds.
> > >
> > > *“experiments would substantially improve the paper”*
> > > \
> > > We reiterate that our work is consistent with other theoretical papers we build on, such as Jin et al. and Li et al., whose focus is mathematical rigorousness. While we agree that experimental results will emphasize the contribution of our method, we believe this is beyond the scope of such a theoretical work and leave it for future research.

---

### Decision · Program_Chairs · 2026-04-30

**Decision:**

Accept (regular)

**Comment:**

Reviewers agree on accepting the article, considering the fix for the proof regarding dependency will indeed likely only modify constant factors and change asymptotic order in the O() notation.

The improvement in the lower-order term, together with the proposed technique, can be considered a meaningful contribution.

I recommend Accept, considering there is enough evidence that the authors will fix this in the camera-ready version.